# Normalize Filters! Classical Wisdom for Deep Vision

**Gustavo Perez**[1]         **Stella X. Yu**[1,2]

[1]Electrical Engineering and Computer Sciences, University of California, Berkeley
[2]Computer Science and Engineering, University of Michigan
{gperezs, stellayu}@berkeley.edu, stellayu@umich.edu

## Abstract

Classical image filters, such as those for averaging or differencing, are carefully normalized to ensure consistency, interpretability, and to avoid artifacts like intensity shifts, halos, or ringing. In contrast, convolutional filters learned end-to-end in deep networks lack such constraints. Although they may resemble wavelets and blob/edge detectors, they are not normalized in the same or any way. Consequently, when images undergo atmospheric transfer, their responses become distorted, leading to incorrect outcomes. We address this limitation by proposing filter normalization, followed by learnable scaling and shifting, akin to batch normalization. This simple yet effective modification ensures that the filters are atmosphere-equivariant, enabling co-domain symmetry. By integrating classical filtering principles into deep learning (applicable to both convolutional neural networks and convolution-dependent vision transformers), our method achieves significant improvements on artificial and natural intensity variation benchmarks. Our ResNet34 could even outperform CLIP by a large margin. Our analysis reveals that unnormalized filters degrade performance, whereas filter normalization regularizes learning, promotes diversity, and improves robustness and generalization.

## 1   Introduction

Image filtering is a cornerstone of classical computer vision; averaging and differencing filters play a fundamental role in noise reduction, edge detection, and feature extraction. These filters are meticulously designed and normalized to ensure consistent and interpretable results (Fig. 1).

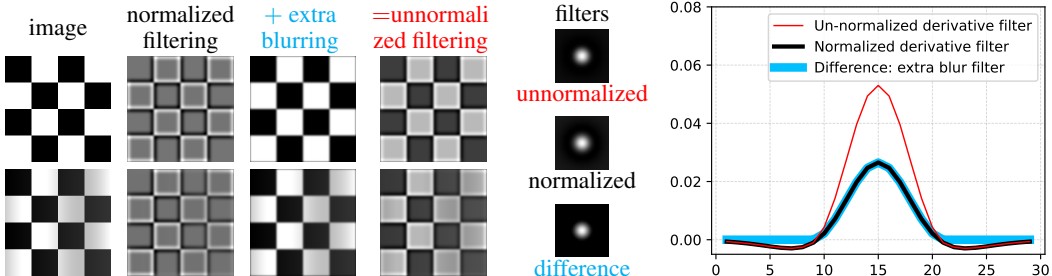

Figure 1: **Filter normalization is critical in classical computer vision for consistency and interpretability, while unnormalized filters obtained by deep learning lack this property. Left:** For a checkerboard (Column 1) under uniform (Row 1) and varied (Row 2) illumination, we compare responses from normalized (Column 2) and unnormalized (Column 4) filters; the latter equals the sum of the former and extra blurring (Column 3). **Right:** For normalized and unnormalized Difference-of-Gaussian (DoG) filters, we show their horizontal profiles at the center row and their difference. Normalized DoG detects edges despite illumination changes, while unnormalized DoG introduces blurring that varies with illumination, overpowering edges with mean response shifts.

39th Conference on Neural Information Processing Systems (NeurIPS 2025).

Filter normalization is critical because it prevents artifacts such as intensity shifts, halos, and ringing, which can distort the output and undermine the reliability of downstream tasks. For example, in Gaussian blurring, the filter kernel is normalized so that its weights sum to 1, preserving the overall intensity of the image while smoothing out fine details. Similarly, Gaussian derivative filters are normalized so that its positive weights sum to $+1$ and negative weights sum to $-1$, enabling accurate edge detection and allowing edge strengths to be compared across different types of derivative filters. This careful normalization process has been a hallmark of classical filtering, ensuring consistency and robustness across a wide range of applications (See Fig. 1).

In contrast, convolutional filters in deep networks are not explicitly designed but are instead learned end-to-end, emerging through an optimization process driven by data and training loss. While these filters may resemble wavelets, blob detectors, or edge detectors [41, 24, 29, 40, 32, 18], they are not normalized in the same - or any - way as their classical counterparts. This lack of normalization can lead to unpredictable behavior, particularly when the input data deviates from the training distribution.

In Fig. 1, for the checkerboard under uniform and uneven illumination, the normalized Difference of Gaussian (DoG) filter consistently detects checker edges. In contrast, the unnormalized DoG filter, whose positive weights sum to $+2$ and negative weights to $-1$, is equivalent to the sum of the normalized DoG and a normalized blur filter. This extra blurring causes the response to shift with illumination and overwhelm the edge response of the normalized DoG.

Such illumination effects, commonly found in autonomous driving, medical imaging, and remote sensing, can be modeled as Atmospheric Transfer Functions (ATFs) [2], which linearly map the physical reflectance of a scene to the luminance in the image. Varying the gain and bias of the linear map simulates real-world effects like brighter/dimmer/hazy scenes (See Fig. 2). However, the responses of unnormalized filters to such intensity changes can become distorted, compromising the neural network's performance. .

Data augmentation and instance normalization [33] are two common methods to address data distribution shifts. However, the former requires anticipating domain shifts in advance and increases training complexity, while the latter assumes that all pixels in an image follow the same statistics and cannot handle spatially varying biases, e.g., Fig. 1.

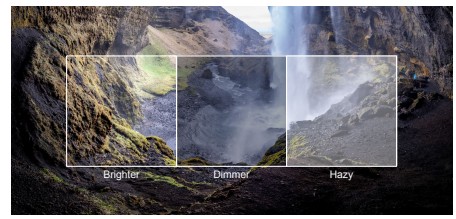

Atmospheric transfer functions

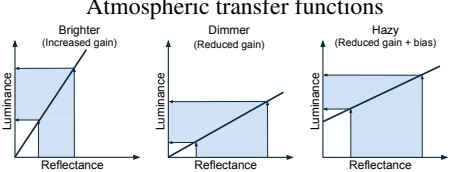

Figure 2: **The same visual scene can be seen very differently under various atmospheric conditions.** Lighting, shadows, and haze can be modeled by Atmospheric Transfer Functions (ATFs) [2], which map the physical reflectance of a scene to the luminance in the image. Varying gains and biases can simulate effects like brighter/dimmer/hazy scenes.

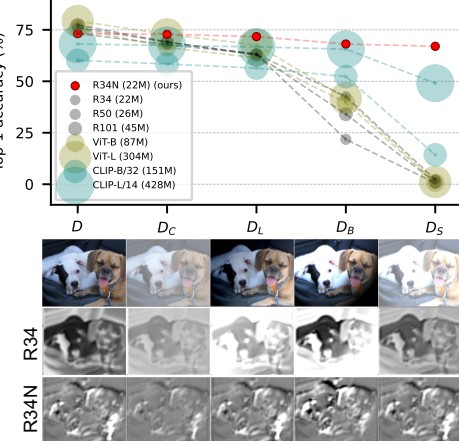

Figure 3: **Feature normalization outperforms larger ResNet and ViT models, including CLIP, in accuracy and robustness on atmospheric transfer benchmarks.** Every model is trained on $D$, the original ImageNet dataset, but tested on $D$ and its four ATF variations: $D_C, D_L, D_B, D_S$, as illustrated for the image example in Row 1. The response of a ResNet34 filter with filter normalization (Row 3) remains more robust and informative than the one without normalization (Row 2) to such intensity transformation.

We propose *filter normalization*, a novel approach that normalizes convolutional filters followed by learnable scaling and shifting, akin to batch normalization. This simple yet effective modification ensures that the filters are atmosphere-equivariant, with consistent responses under varying conditions.

Our extensive experiments on artificial and natural intensity variation benchmarks demonstrate that our approach is not only robust but also outperforms larger models of ResNet [13] and Vision Transformer (ViT) [10] architectures. Most strikingly, ResNet34 with feature normalization surpasses CLIP, the

extensively trained large vision-language model, by an absolute 20% on ImageNet in hazy conditions (Fig. 3). Consistent gains are observed on ImageNet, ExDark, and LEGUS benchmarks. Our analysis further reveals that unnormalized filters degrade performance, whereas filter normalization regularizes learning and promotes diversity, leading to better robustness and generalization.

**Our contributions. 1)** Identifies unnormalized filters as a key limitation in deep learning, distorting responses under intensity variations. **2)** Proposes *filter normalization* with learnable scaling/shifting, ensuring *atmosphere equivariance*. **3)** Achieves consistent gains across CNNs and ViTs on intensity variation benchmarks, outperforming larger models, including CLIP. **4)** Enhances robustness and generalization through diverse, interpretable filters.

## 2   Related work

Deep networks' performance can be severely impacted by atmospheric transformations (we use "atmospheric" in the context of *"The net effect of the viewing conditions, including additive and multiplicative effects"* [2]). To mitigate this, various approaches have been developed such as architecture modifications to achieve equivariance to some transformations, normalization layers to enhance network robustness to instance-level perturbations, and data augmentations to improve robustness and generalization.

**Equivariance and invariance to transformations.**   Several works have proposed modifications to deep neural networks to achieve equivariance or invariance to specific transformations. For instance, [5, 22, 19] introduced model modifications to achieve equivariance to rotations, [36] used circular harmonics to achieve equivariance to rotation and translation, [11] achieved invariance to translation and equivariance to rotation and scale, while [23, 15] focus on image denoising.

Complex-valued deep learning has also been studied to achieve codomain symmetry to geometric and color transformations [30]. [4] is another approach that achieves equivariance through transformations on a scaling and rotation manifold. To the best of our knowledge, other than [15, 6], existing methods focus on geometric transformations and overlook atmospheric variations like gain-bias shifts. Closest to our work, [15] proposes equivariant networks to gain and bias. However, our proposed normalization also provides practical invariance to bias, leading to improved robustness.

**Normalization layers.**   Instance Normalization (IN) [33] layers can be added to the network to reduce instance-specific biases, improving robustness to multiplicative and additive effects. However, IN may not fully address covariate shift, leading to suboptimal training. On the other hand, batch normalization (BN) [16] layers reduce covariate shift, but fail to address instance-specific biases. Other normalization techniques, such as group normalization (GN) [38], which balances the benefits of BN and IN by dividing channels into groups, have also been proposed. However, these methods operate on feature activations, whereas our approach introduces a data-agnostic modification to the convolutional layer by normalizing filter weights. This allows our method to address instance-specific biases while still enabling training with BN to handle covariate shift, resulting in improved training.

**Data augmentation.**   Atmospheric effects are frequently addressed with data augmentations, which are a common technique for improving model robustness. Automated augmentation strategies like RandAugment [8] and AutoAugment [7] have been proposed to reduce the search space of data perturbations. However, they increase training complexity and require prior knowledge of the target domain since it is challenging to account for all possible variations in the deployment data. In contrast, our approach provides inherent robustness to these perturbations, eliminating the need for extensive data augmentation.

**Deep network architectures.**   CNN architectures such as ResNets [13] have been successful due to their ability to capture spatial hierarchies and local correlations. ViTs [10], on the other hand, employ an attention-based mechanism that allows for global interactions between image regions, enabling more flexible and context-dependent feature extraction. However, more recent architectures like ConvNeXt [20] have demonstrated comparable performance to ViTs, while maintaining the efficiency and interpretability of traditional CNNs. Other approaches like Convolutional Vision Transformer (CvT) [37] improve ViTs by adding convolutions. Our approach can be applied to any convolution-based architecture, and we show that leaner models with our normalized convolutions outperform large attention-based models like CLIP [26] in robustness to atmospheric corruptions.

## 3 Filter normalization for deep learning

We have shown that unnormalized filters obtained by deep learning lack consistency and interpretability with respect to atmospheric transfer functions (Fig. 1). We will first relate an arbitrary filter to normalized filters followed by scaling and shifting. We then prove atmosphere equivariance of normalized filters.

### 3.1 Any linear filter is both averaging and differencing

We connect a convolutional filter with weights $w$ to classical averaging (blur) and differencing (derivative) filters. The filter response, $y$, to $k$ inputs $x_1, \ldots, x_k$, is defined as:

$$y = f(x; w) = \sum_{i=1}^{k} w_i x_i. \tag{1}$$

We divide $w$ into positive and negative parts. Let $1(\cdot)$ denote the indicator function which outputs 1 if the input is positive and 0 otherwise. Let $\circ$ denote the Hadamard product (multiplication of elements). We have:

$$w = w^+ - w^- \tag{2}$$

$$w^+ = w \circ 1(w > 0) \tag{3}$$

$$w^- = (-w) \circ 1(w < 0) \tag{4}$$

where both $w^+$ and $w^-$ are nonnegative $k \times 1$ vectors. The $L_1$ norm $\| \cdot \|_1$ of these vectors is simply the total sum of their respective weights. We will use $L_1$ norm exclusively throughout this paper, with $\| \cdot \|_1$ as $\| \cdot \|$ for short.

We define the *positive weight ratio* of a filter, denoted by $r$, as the ratio between the algebraic sum and the total absolute sum of all the weights (See Fig. 4). We have:

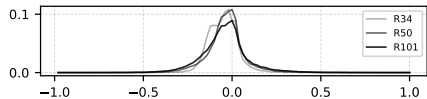

$$r(w) = \frac{\sum_{i=1}^{k} w_i}{\sum_{i=1}^{k} |w_i|} = \frac{\|w^+\| - \|w^-\|}{\|w^+\| + \|w^-\|} \tag{5}$$

$$\sum_{i=1}^{k} w_i = \|w^+\| - \|w^-\| = \|w\| \cdot r(w). \tag{6}$$

It is straightforward to prove the following properties.

**Figure 4: Convolutional filters learned in CNNs are unnormalized, leaning more toward differencing than averaging.** The distribution of the *positive weight ratio* of filters for each model optimized for ImageNet classification is skewed towards 0, indicating that most filters primarily perform differencing.

1. $|r(w)| \leq 1$.

2. $r(w) = 1$ (–1) when and only when all non-zero weights of $w$ are positive (negative).

3. $r(w) \gtreqless 0$ when and only when $\|w^+\| \gtreqless \|w^-\|$.

In classical computer vision, there are two types of filters: averaging and differencing filters, each type normalized properly to ensure consistency and interpretability:

$$\text{averaging: } \|w^+\| = 1, \|w^-\| = 0, r(w) = 1 \tag{7}$$

$$\text{differencing: } \|w^+\| = 1, \|w^-\| = 1, r(w) = 0 \tag{8}$$

That is, an averaging filter produces a normalized weighted average of its inputs, and a differencing filter produces the difference between two normalized weighted sums, one using positive weights and the other negative weights.

We show that an unnormalized filter $w$ is a weighted sum of a differencing filter and an averaging filter (Fig. 1). Without loss of generality, we assume $\|w^+\| \geq \|w^-\|$; otherwise, we can study $-w$ instead. We have:

$$w = w^+ - w^- = \|w^-\| \cdot \underbrace{\left( \frac{w^+}{\|w^+\|} - \frac{w^-}{\|w^-\|} \right)}_{\text{differencing filter}} + (\|w^+\| - \|w^-\|) \cdot \underbrace{\frac{w^+}{\|w^+\|}}_{\text{averaging filter}}. \tag{9}$$

Therefore, a general filter behaves between averaging and differencing, and the closer its positive weight ratio is to 0 (1), the more dominant differencing (averaging) it becomes.

## 3.2 Filter normalization

Consider the input intensity $x$ modulated by a scalar gain factor $g$ and a scalar offset $o$:

$$x \rightarrow g \cdot x + o. \tag{10}$$

If $g > 1$ ($g < 1$), the image looks brighter (dimmer) with an expanded (reduced) range; if $g < 1$ and $o > 0$, the image looks brighter and hazy with reduced contrast (Fig. 2). For any unnormalized filter $w$, the response becomes

$$f(g \cdot x + o) = g \cdot f(x) + o \cdot \|w\| \cdot r(w). \tag{11}$$

If all filters in a layer are unnormalized, each response is scaled by the filter's total weight sum, $\|w\| r(w)$, thereby altering the feature representation at that layer. However, if all filters are normalized, their responses are scaled uniformly by the same gain factor $g$, that is, they produce consistent and interpretable responses with respect to intensity changes.

We address this limitation by proposing filter normalization, followed by learnable scaling and shifting, akin to batch normalization. A convolutional filter of kernel size $k$ in deep learning is parametrized by weight $w$ and scalar offset $b$. Given an arbitrary filter $w$, we first normalize its positive and negative parts individually:

$$y = \sum_{i=1}^{k} w_i x_i + b = \|w\| \sum_{i=1}^{k} \frac{w_i}{\|w\|} x_i + b = \|w\| \sum_{i=1}^{k} \left( \frac{w_i^+}{\|w\|} - \frac{w_i^-}{\|w\|} \right) x_i + b \tag{12}$$

$$\rightarrow \underbrace{a}_{\text{scaling}} \sum_{i=1}^{k} \underbrace{\left( \frac{w_i^+}{\|w^+\| + \varepsilon} - \frac{w_i^-}{\|w^-\| + \varepsilon} \right) x_i}_{\text{filter normalization}} + \underbrace{b}_{\text{shifting}}, \tag{13}$$

where $\varepsilon$ is a small constant (e.g., $10^{-6}$) to ensure numerical stability. This is followed by learnable scaling $a$ and shifting $b$, which model response weighting $\|w\|$ and offset $b$, respectively.

This normalization step enforces the filter to become *either* averaging if $w$ is all positive, *or* differencing if $w$ has both positive and negative weights:

1. Averaging filters are equivariant to both gain and offset: $f(g \cdot x + o) = g \cdot f(x) + o$, where $\|w\| = 1$ and $r(w) = 1$.

2. Differencing filters are invariant to offset and equivariant to gain: $f(g \cdot x + o) = g \cdot f(x)$, where $\|w\| = 1$ and $r(w) = 0$.

This decomposition ensures atmosphere-equivariance and enable co-domain symmetry, meaning the filter response transforms in a predictable, structured way under affine transformations of the input intensity $x$, via gain $g$ and offset $o$, (i.e., global illumination changes.) Averaging filters respond to both contrast and brightness, while differencing filters respond only to contrast.

## 4 Experiments

Our approach normalizes convolution filters in deep networks to eliminate artifacts and enhance robustness to atmospheric perturbations. While we demonstrate its effectiveness on ResNets [13], the method is broadly applicable to any model with convolutional layers. We evaluate our approach on artificially corrupted ImageNet [9] and CIFAR [17] datasets for image classification tasks, as well as on natural data with intensity variations, including low-light and astronomy datasets. Additionally, we perform extensive analyses, ablations, and comparisons with alternative techniques such as data augmentation and normalization layers.

### 4.1 Results on artificially corrupted benchmarks

We use the original ImageNet-1k [9] and CIFAR-10 [17] training sets to train our models. ImageNet-1k contains over a million training images, 50K validation images, and 1,000 classes. CIFAR-10 has 10 categories, with 50K and 10K training and testing images respectively. We denote the original evaluation sets of these datasets as $D$.

We propose four additional evaluation sets by applying random gain and bias perturbations to $D$. **1)** $D_C$ has *constant* random gain-bias corruptions to the entire image: $\tilde{x} = \alpha x + \beta$, where $\tilde{x}$ is the

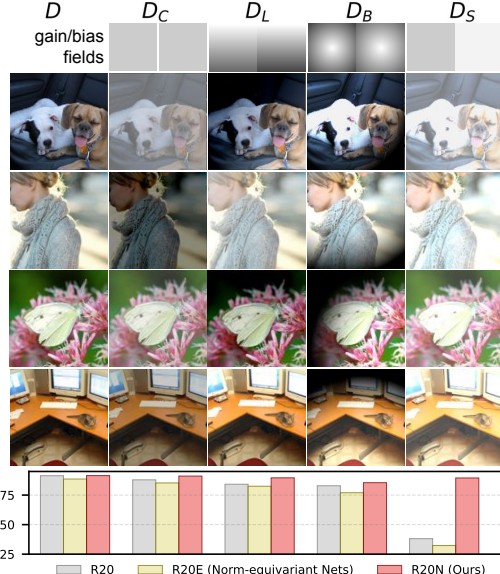

Figure 5: **Top: our corrupted evaluation benchmarks simulate realistic scenarios with varying global and local illumination sources and atmospheric effects.** We add corruptions to ImageNet original validation set ($D$). The first three ($D_{C,L,B}$) sets apply random atmospheres (See Fig. 2) in a constant, linear, and blob fashion. $D_S$ applies a constant random variation with a fixed shift. **Bottom: our approach provides more robust results on the corrupted datasets and surpasses vanilla and normalization-equivariant networks across architectures while maintaining performance on the original test set.** We train vanilla (R20) and normalization-equivariant nets (R20E) [15] on the original CIFAR-10 training set and show classification accuracy on $D, D_C, D_L, D_B$, and $D_S$. Our approach (R20N) achieves a significantly lower accuracy drop compared to vanilla R20 and norm-equivariant R20E. See complete CIFAR-10 results in Appendix A.1.

corrupted image, and $\alpha \sim \text{Unif}(0.7, 1.3)$ and $\beta \sim \text{Unif}(-0.3, 0.3)$ are calculated to have a maximum variation of $\pm 30\%$ from the original image $x$ intensity. **2)** $D_L$ has gain-bias corruption in a smooth *linearly* decreasing fashion: $\tilde{x} = x \odot \mathbf{L}_\alpha + \mathbf{L}_\beta$, where $\odot$ denotes the Hadamard product, $\mathbf{L}_\alpha$ is a linearly varying field with values $[\alpha_0, \alpha_1]$ where $\alpha_0, \alpha_1 \sim \text{Unif}(0.5, 1.5)$, and $\mathbf{L}_\beta$ a different linear field with values $[\beta_0, \beta_1]$ where $\beta_0, \beta_1 \sim \text{Unif}(-0.5, 0.5)$, both fields with the same random direction. **3)** $D_B$ has the gain-bias perturbation in a *blob* with a fixed but decaying radius centered on a random pixel in the image: $\tilde{x} = x \odot \mathbf{B}_\alpha + \mathbf{B}_\beta$, where $\mathbf{B}_\alpha$ and $\mathbf{B}_\beta$ are blob fields calculated with a cubic decay over a radius of 0.8 the size of the image. $D_C, D_L, D_B$ simulate global and local atmospheric effects such as variations in illumination caused by the environment or artificial light sources, calibration, or artifacts from sensor defects (See Fig. 2). **4)** $D_S$ perturbation is also constant as in $D_C$, but with a fixed strong *shifted* bias to increase the data distribution shift: $\tilde{x} = \mathbf{1}(\alpha x + \beta + \gamma)$, where $\alpha \sim \text{Unif}(0.7, 1.3)$, $\beta \sim \text{Unif}(-0.3, 0.3)$ and $\gamma = 1$.

We anticipate that our four corrupted evaluation sets will exhibit increasing levels of difficulty; $D_C$ introduces a uniform perturbation across the entire image, $D_L$ applies a linearly varying gain and bias, while $D_B$ imposes a perturbation with a cubic decay from a randomly selected set of coordinates. On the other hand, $D_S$ also applies a uniform corruption, but with an additional fixed bias shift, which further increases the divergence between the testing data distribution and the original training set.

**Our normalized convolutions are more robust across architectures on CIFAR-10.** We evaluate our normalized convolutions (R20N) on CIFAR with vanilla ResNets [13] (R20). Additionally, we compare to norm-equivariant nets [15] (R20E), which is the most recent baseline and closest related to our method. Norm-equivariant networks replace ordinary convolutional layers with affine-constrained convolutions and traditional activation functions like ReLU with channel-wise sort pooling layers to preserve normalization-equivariance by design, allowing the network to handle changes in input scale and shift. Following their official implementation [15], we replace the vanilla convolutions and activations in the ResNets with affine convolutions and sort pooling layers, respectively.

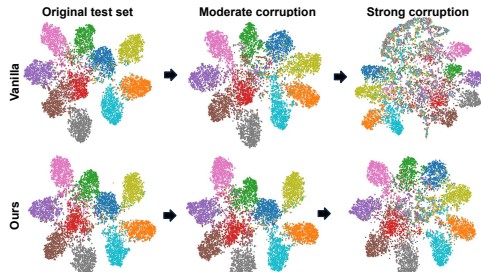

Figure 6: **Feature quality from unnormalized convolutions drops under atmospheric perturbations.** We show the t-SNE visualization of a R20 last layer's features with moderate and strong levels of corruption on CIFAR-10. **Top:** A vanilla R20 fails under atmospheric perturbations ($D_c$), while **Bottom:** our approach maintains the feature quality even with strong perturbations ($D_C$ with maximum variation of $\pm 100\%$).

We train the three models (R20, R20E, R20N) using SGD with a learning rate of 0.1, a cosine annealing schedule for 200 epochs, and a batch size of 128. Fig. 5–*bottom* shows our approach's

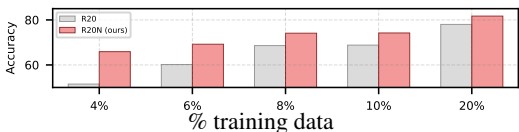

Figure 7: **Low-shot training benefits from normalized convolutions.** We show the accuracy on the $D_C$ set when training with fewer labeled images (indicated by %). Our approach generates more robust features against atmospheric effects, while also avoiding artifacts that can occur with unnormalized convolutions.

Table 1: **Our approach surpasses vanilla CNNs and ViT-based architectures on the corrupted ImageNet-1k.** We show Top-1 accuracy on the original validation set ($D$) and our proposed benchmarks ($D_C, D_L, D_B, D_S$) for various pretrained ResNets, ViTs, and CLIP (LAION). Our method provides more robust results on the corrupted datasets while maintaining close performance on the original set ($D$). Notably, R34N (22M params.) outperforms larger models like ResNet101, ViT-large, and CLIP.

| Model | # par (M) | $D$ | $D_C$ | $D_L$ | $D_B$ | $D_S$ |
|---|---|---|---|---|---|---|
| R34 [13] | 22 | 73.3 | 68.6 | 64.5 | 36.8 | 2.1 |
| R50 [13] | 26 | 76.1 | 69.0 | 63.1 | 21.8 | 0.6 |
| R101 [13] | 45 | 77.4 | 69.1 | 62.9 | 34.0 | 1.1 |
| ViT-B [10] | 87 | 75.7 | 66.5 | 61.3 | 39.6 | 0.7 |
| ViT-L [10] | 304 | **79.3** | **72.8** | 67.7 | 41.8 | 1.1 |
| CLIP-B/32 [26]† | 151 | 60.2 | 58.4 | 56.4 | 52.3 | 14.3 |
| CLIP-L/14 [26]† | 428 | 68.1 | 67.4 | 66.7 | 65.6 | 49.1 |
| R34N (ours) | 22 | 73.2 | **72.8** | **71.7** | **68.1** | **67.0** |

†Trained on LAION-1B (Zero-shot classification accuracy)

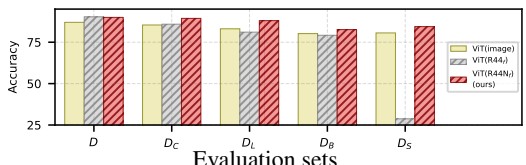

Figure 8: **Features from our normalized convolutions improve downstream classification on ViTs.** We show the accuracy of small ViTs on CIFAR-10 when trained using feature maps from a R44 as input. ViTs are more robust than vanilla ResNets, but using feature maps from our normalized convolutions reduces the performance drop against atmospheric effects ($D_C, D_L, D_B, D_S$).

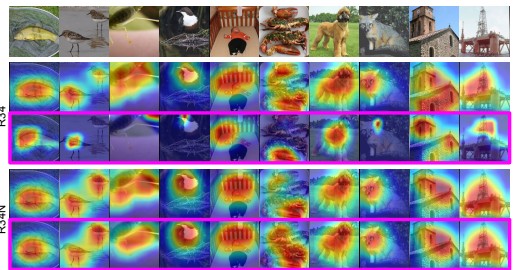

Figure 9: **Our convolutions maintain their focus on salient regions despite the presence of atmospheric effects**. Grad-CAM visualizations show that a vanilla R34 trained on ImageNet loses focus on relevant regions under strong corruptions (top), while our R34N produces more stable feature representations (bottom).

superior robustness to atmospheric effects, outperforming vanilla R20 and norm-equivariant R20E. See complete results with larger models in Table A1.

To further analyze the impact of corruptions on feature representations, we visualize the last layer features on CIFAR-10's test set using t-SNE [34]. As shown in Fig. 6–*top*, a vanilla R20 struggles to produce discriminative features in the presence of atmospheric variations ($D_C$ ), whereas our approach (Fig. 6–*bottom*) preserves features more effectively even under severe corruptions ($D_C$ with max. variation of $\pm100\%$). In Appendix A.1.1 we show the performance of R20 and R20N on $D_C$ varying the amount of corruption.

**Low-shot learning benefits from normalized convolutions.** We investigate the benefits of our approach to other tasks. Our proposed convolutions not only produce equivariant features to gain and offset variations but also reduce the number of artifacts produced by unnormalized contrast filters. We expect these features to be particularly advantageous in low-data regimes. Table A2 shows our normalized convolutions outperforming on the original evaluation set $D$ for very low data regimes. Additionally, Fig. 7 and Table A3 show the effectiveness of our approach on $D_C$, outperforming a vanilla R20 when trained with limited data. Notably, the performance gain between our approach and the baseline widens as the amount of labeled data decreases (14.4% difference with 4% data).

**Features from our normalized convolutions improve downstream classification on ViTs.** For this set of experiments we train a small ViT [39] on CIFAR-10 with $4 \times 4$ patch size, 6 layers, 8 heads, and training hyperparameters matching those in ResNet experiments. Fig. 8 and Table A4 show that, although vulnerable to atmospheric effects, ViTs exhibit greater robustness than vanilla ResNets in this task. For instance, the small ViT achieves 80.6% accuracy on the $D_S$ benchmark, outperforming a vanilla R44.

Next, we experiment using ResNet feature maps as input to the ViT. We use the feature maps of the last convolutional layer ($16 \times 16$) as input to the ViT with the same $4 \times 4$ patch size. In Fig. 8 and Table A4 we show that using feature maps from our normalized convolutions (R44N$_f$) mitigates the impact of perturbations, outperforming the original ViT by 3.8% (84.4% vs. 80.6%) on $D_S$

Table 2: **Accuracy per contrast level.** We bin the ImageNet-1k test set into 9 contrast levels (based on pixel standard deviation). R34N maintains more stable accuracy across the contrast spectrum compared to the baseline, which drops sharply at the extreme contrast values. **Top row:** example images from the uncorrupted test set for the lowest-contrast bin (left), the middle bin (center), and the highest-contrast bin (right). **Bottom:** accuracy table showing performance for each contrast bin.

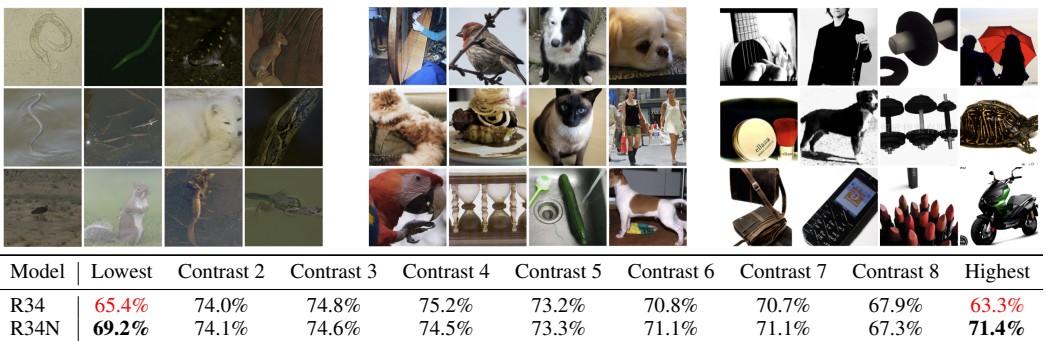

| Model | Lowest | Contrast 2 | Contrast 3 | Contrast 4 | Contrast 5 | Contrast 6 | Contrast 7 | Contrast 8 | Highest |
|-------|--------|-----------|-----------|-----------|-----------|-----------|-----------|-----------|---------|
| R34 | 65.4% | 74.0% | 74.8% | 75.2% | 73.2% | 70.8% | 70.7% | 67.9% | 63.3% |
| R34N | **69.2%** | 74.1% | 74.6% | 74.5% | 73.3% | 71.1% | 71.1% | 67.3% | **71.4%** |

**Our approach achieves consistent gains across CNNs and ViTs on ImageNet.** We train a R34 with our normalized convolutions from scratch on the original ImageNet-1k training set for 90 epochs, a batch size of 256, and SGD with an initial learning rate of 0.1 divided by 10 every 30 epochs. In Table 1 we show that our approach (R34N) is more resilient to atmospheric corruptions, with up to 5% performance drops on our proposed benchmarks ($D_C, D_L, D_B, D_S$), whereas a vanilla R34 suffers significant drops of up to 97% on the most corrupted datasets. Additionally, Grad-CAM [28] visualizations in Fig. 9 show that R34 loses focus from salient regions under corruptions while our R34N produces more stable representations.

Notably, our R34N with 22M parameters achieves better results than larger models like ResNet101 (45M), ViT-B (87M), and CLIP [26] (151M) trained on much larger datasets like LAION [27]. Although CLIP is used here as a zero-shot classifier and our R34N is trained specifically for ImageNet classification, the comparison is fair and meaningful because both models face the same unseen corruptions at test time. In fact, the comparison disadvantages our method, as CLIP was trained on 400M diverse images and likely encountered many degradations that our model has not seen. Despite this, R34N outperforms CLIP on intensity corruptions, highlighting the effectiveness of our filter normalization approach. On the original test set $D$, specialist models like R34N and R34 outperform generalist models like CLIP-B (∼73% vs. 60.2%), reflecting their domain-specific focus. However, on corrupted sets, specialist baselines collapse (e.g., R34 drops from 73.3% to 2.1%), while CLIP-B drops but to a higher accuracy (from 60% to 14.3%). In contrast, R34N achieves 67.0% on corrupted sets, far surpassing both R34 and CLIP-B. This demonstrates that the robustness gain comes from the architecture itself, not from exposure to corruptions or tradeoffs on original clean performance.

**Robustness on original ImageNet images across low and high contrast.** To evaluate robustness to natural variation, we partition the uncorrupted ImageNet-1k test set into nine contrast bins defined by pixel-level standard deviation. As shown in Table 2, R34N exhibits notably more consistent accuracy across contrast levels, while the baseline model suffers substantial degradation at the extremes. These results highlight the improved robustness of R34N to natural, realistic atmospheric conditions.

## 4.2 Results on natural data

**Our normalized convolutions are more robust under extreme low light conditions on ExDark dataset.** We evaluate our approach on the Exclusively Dark (ExDark) Image Dataset [21], which con-

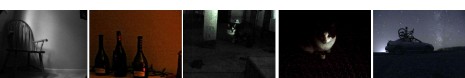

Figure 10: Samples from the ExDark dataset.

tains over 7,000 images captured in 10 different low-light conditions. We focus on the images labeled as "low light" (Fig. 10), to validate the robustness of our approach to extreme low-light scenes. The ExDark dataset provides image-level labels for 10 classes, which are related to ImageNet categories. Specifically, the ExDark classes are a coarser version of the ImageNet categories (e.g., ExDark has a "dog" class, while ImageNet has multiple classes for different dog species). We do a zero-shot evaluation on ExDark using a vanilla R34 and our R34N, both trained on ImageNet. We map the ImageNet labels to the coarser ExDark labels and evaluate the models. Our R34N achieves 34.2% classification accuracy, outperforming R34 (28.3%).

**Our normalized convolutions generalize better across galaxies in astronomy data from LEGUS.** Star cluster classification from galaxies is an active research area in astrophysics, as it provides insights into the process of star formation [12, 1]. As new telescopes like the James Webb (JWST) continue to capture new data, models that can generalize across galaxies becomes increasingly important.

We explore the potential of our normalized convolutions using the LEGUS dataset [3], which contains data from 34 galaxies captured by the Hubble telescope (HST) with 5 spectral channels and ∼15,000 annotated star clusters and other sources. We evaluate the robustness to varying luminance conditions by training and evaluating on different targets. Specifically, we use galaxies NGC628 and NGC1313 which have a noticeable difference in intensity (Fig. 11a).

We train a R18 on NGC628 using around 2,000 object-centered patches of size $32 \times 32 \times 5$ from coordinates provided in LEGUS (See Fig. 11b), to classify them into one of four morphological classes. We use training hyperparameters as in [25] and evaluate on galaxy NGC1313 sources. Our results show an accuracy of 50.2% with vanilla R18, while our R18N improves accuracy to 51.9%.

**a)** NGC628 (left) and NGC1313 (right) from HST

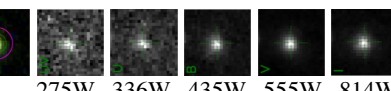

**b)** A single star cluster with its 5 spectral bands

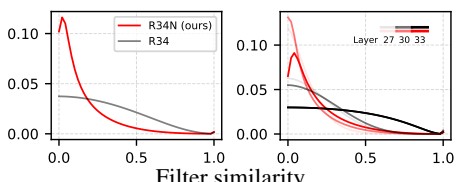

275W  336W  435W  555W  814W

Figure 11: **Astronomy data suffers from significant intensity variations, similar to atmospheric effects on standard photographs. a)** Galaxies NGC628 (left) and NGC1313 (right) from HST data, showing higher luminance in NGC1313 due to background light caused by dust and gas. **b)** A star cluster crop with its five spectral bands (275W-814W) from the LEGUS dataset.

### 4.3 Analysis and ablation studies

**Unnormalized filters learn less diverse representations.** Normalization can promote diversity by forcing the filters to lie on a specific manifold. We study filter similarity using guided backpropagation [31] to visualize filter gradients with respect to the images on the ImageNet validation set. Then, we compute the correlation matrix of the guided backpropagation maps of each filter in a layer and plot the histogram of its off-diagonal elements. Low values from the off-diagonal elements indicate less similar features.

Fig. 12 shows that our R34N learns less correlated features, with a histogram more skewed towards zero.

**Unnormalized filters produce more errors.** We investigate the relationship between a filter's level of "unnormalization" and misclassifications. First, we calculate the positive weight ratio ($|r(w)|$) of the

Figure 12: **Unnormalized filters produce less diverse features.** We compare the filter similarity histograms of a vanilla R34) and our R34N trained on ImageNet. **Left:** The histogram for the last eight convolutional layers shows that R34N has a more skewed distribution towards lower similarity values than vanilla R34. **Right:** Consistent with [35], layer-wise histograms reveal that vanilla R34 produces more similar features in later layers, whereas R34N does not exhibit this trend.

filters in the first layer of an ImageNet pretrained R34. We expect this metric to be 0 for normalized differencing filters (See Fig. 13–*left*). Then, focusing on the first layer, where gain and bias effects are largest, we find the Top 100 images on $D_C$ that maximize the response for each filter and calculate the number of misclassified ones by the network.

In Fig. 13–*right* we show that for a R34, the images with a higher response from filters with higher $|r(w)|$ are more likely to be misclassified. For instance, filters with highest $|r(w)|$ misclassify ∼67% of the images, while the least unnormalized misclassify ∼30%. Our R34N avoid this issue by design.

**Are data augmentations enough?** To evaluate the effectiveness of data augmentation, we train a vanilla R20 and our proposed R20N using atmospheric augmentations on CIFAR-10. We use random gain-bias augmentations with a maximum variation of $\pm10\%$ from the original image intensity to simulate the case where we want to approximate the testing data distribution but we do not know the *exact* distribution (e.g., $\pm30\%$ maximum variation on $D_C$).

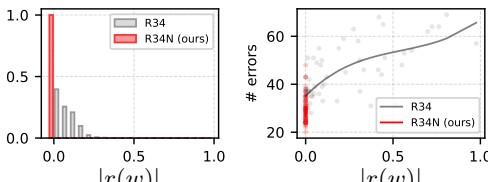 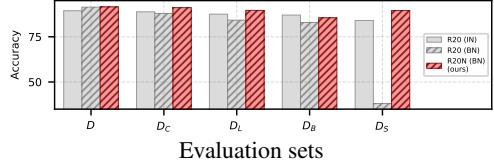

Figure 13: **Unnormalized filters misclassify more often.** We investigate how filter weight normalization affects error rates in a pretrained R34. **Left:** We show the histogram of $|r(w)|$ from all filter weights in the network. We want this ratio to be 0 if the filters are differencing, or 1 if they are averaging filters. The peak in 0 from our R34N indicates all our filters are differencing and normalized, while vanilla ResNets possess many unnormalized. **Right:** We find that images triggering strong responses from "less normalized" filters (i.e., higher $|r(w)|$) are more often misclassified. Our convolutions avoid this by design.

Figure 14: **Instance normalization improves robustness to atmospheric effects but may not fully address covariate shift, leading to suboptimal training.** We compare a vanilla R44 trained on CIFAR-10 with instance normalization layers to a vanilla R44 with batch normalization and our R44N. We get best results on most benchmarks while maintaining original test set accuracy ($D$). Our approach differs from instance norm. in two ways: independent normalization of positive and negative weights, reducing artifacts; and normalizing weights, not activations.

Our results in Table 3 show that data augmentation provides significant robustness, even with lower gain-bias variation than the test set. However, when data distributions differ substantially, as in $D_S$, data augmentation falls short, with our R20N achieving 91% accuracy vs. 77% from R20.

Table 3: **Training with data augmentations helps but struggles with significantly different data distributions.** We train a R20 with and without gain-bias augmentations and evaluate on our proposed benchmarks. **Left:** Without needing gain-bias augmentations, our R20N beats vanilla R20 in all corrupted datasets. **Right:** While using augmentations improves accuracy across all benchmarks for vanilla R20, it struggles to generalize to significantly different data distributions, such as $D_S$.

| Without data augmentations | | | | | |
|---|---|---|---|---|---|
| Model | $D$ | $D_C$ | $D_L$ | $D_B$ | $D_S$ |
| R20 | 91.4 | 87.9 | 84.2 | 82.9 | 38.1 |
| R20N (ours) | **91.5** | **91.1** | **89.6** | **85.5** | **89.5** |

| With data augmentations | | | | | |
|---|---|---|---|---|---|
| Model | $D$ | $D_C$ | $D_L$ | $D_B$ | $D_S$ |
| R20 | **91.9** | 91.6 | **90.8** | **88.0** | 76.9 |
| R20N (ours) | **91.9** | **91.7** | **90.8** | 87.1 | **90.8** |

**Comparison to instance normalization (IN).** IN reduces instance-specific biases, improving robustness against atmospheric variations as shown in Fig. 14. In contrast, a R20 with BN suffers significant accuracy drops ($\sim$60%). However, IN may not fully address covariate shift, leading to suboptimal training (89.0% vs. 91.4% accuracy on the original set with R20+BN). BN stabilizes training by addressing internal covariate shift but does not improve codomain symmetry like IN and our approach. Our method lets us use BN while taking care of the atmospheric effects, achieving improved performance on most benchmarks while maintaining accuracy on $D$.

While both IN and our approach mitigate instance-specific biases, ours offers two key distinctions: 1) independent normalization of weights, which reduces artifacts that can occur when instance statistics are not representative of atmospheric corruptions; and 2) normalization of weights rather than activations, providing a more targeted correction by directly correcting the filter's response to atmospheric corruptions, rather than indirectly adjusting the activations.

**Negligible computational overhead.** Filter normalization's potential expressivity loss is addressed by learnable scale and shift parameters, adding only 0.08% parameters to an R34. Notably, this increase is unnecessary with batch normalization since these parameters are learned in the batchnorm layers. Our approach also incurs almost no inference-time overhead; only a +0.06 ms per image increase for R34, when measured on ImageNet's validation set using an RTX 2080 Ti.

**Summary.** We identify unnormalized filters as a key deep learning limitation and propose filter normalization, ensuring atmospheric equivariance. We achieve consistent gains across CNNs and ViTs, outperforming larger models like CLIP, while enhancing robustness and generalization. Code available at: https://github.com/gperezs/normalized-convolutions.

## Acknowledgments

This project was supported, in part, by NSF 2215542, NSF 2313151, and Bosch gift funds to S. Yu at UC Berkeley and the University of Michigan, with additional compute support provided by the NAIRR Pilot under CIS240431.

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

# Normalize Filters! Classical Wisdom for Deep Vision
## Supplementary material

## Table of Contents

# A Supporting experiments

## A.1 Classification on CIFAR-10

Table A1: **Accuracy on CIFAR-10.** We train the models on the original training sets and evaluate them on the original test set ($D$) and our proposed benchmarks ($D_{C,L,B,S}$). Our method provides more robust results with all models on the corrupted datasets while maintaining the same performance on the original test set.

|  | Model (vanilla [13] / norm-equiv [15] / ours) | | | | | | | | |
|---|---|---|---|---|---|---|---|---|---|
|  | R20 | R20E | R20N | R32 | R32E | R32N | R44 | R44E | R44N |
| $D$ | 91.4 | 88.6 | **91.5** | **92.0** | 89.7 | **92.0** | **93.2** | 89.0 | **93.2** |
| $D_C$ | 87.9 | 85.3 | **91.1** | 87.7 | 86.5 | **91.5** | 89.7 | 86.7 | **92.6** |
| $D_L$ | 84.2 | 82.5 | **89.6** | 84.3 | 84.0 | **90.4** | 86.4 | 83.6 | **92.0** |
| $D_B$ | 82.9 | 77.0 | **85.5** | 80.1 | 76.2 | **87.6** | 87.3 | 80.6 | **87.8** |
| $D_S$ | 38.1 | 32.3 | **89.5** | 28.6 | 27.8 | **88.8** | 49.5 | 52.2 | **90.2** |

### A.1.1 Ablation with increasing corruption

Here we show the accuracy of a vanilla R20 and our proposed R20N evaluated on the corrupted CIFAR-10 evaluation set $D_C$ with varying amounts of corruption. In Fig. A1 we can see that our R20N is more robust to increasing amounts of gain and bias perturbations.

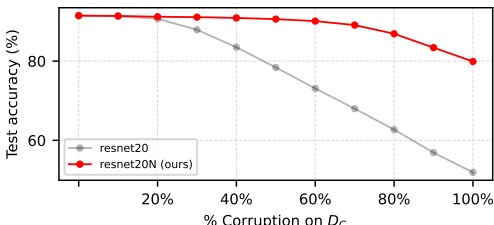

Figure A1: **Our normalized convolutions are more robust to increasing levels of corruption.** A vanilla R20 and our proposed R20N trained on the original CIFAR-10 training set and evaluated on the corrupted evaluation set $D_C$ with increasing levels of gain and bias corruption.

## A.2 Low-shot learning

Table A2 shows our normalized convolutions outperforming on the original evaluation set $D$ for very low data regimes. In Table. A3 we show results for low-show training experiments. Our approach suffers less from reduced training data. Notably, the performance gap between our approach and the baseline widens as the amount of labeled data decreases. See § 4.1 for more details.

Table A2: **Low-shot training results on $D$.** Accuracy on the $D$ evaluation set in very low data regimes (indicated by %). Our approach suffers less from reduced training data.

| % train data | 1% | 2% | 4% | 6% | 8% | 10% |
|---|---|---|---|---|---|---|
| R20 | 41.4 | 51.0 | 61.6 | 68.0 | 74.0 | 75.4 |
| R20N (ours) | **46.3** | **55.8** | **67.0** | **69.9** | **75.0** | **75.6** |

## A.3 Downstream classification with ViTs

In Table. A4 we show our experiments on downstream classification with ViTs using R20, R32, and R44 features as input. We compare to our features using normalized convolutions. See § 4.1 for more details.

## A.4 Comparison to normalization layers

In Table. A5 we present the results with different normalization layers for R20, R32, and R44 on CIFAR-10. Specifically, we show results with instance normalization (IN), batch normalization (BN),

Table A3: **Low-shot training results on** $D_C$**.** Accuracy on the $D_C$ evaluation set when training with fewer labeled images (indicated by %). Our approach suffers less from reduced training data. Notably, the performance gap between our approach and the baseline widens as the amount of labeled data decreases (bottom row).

| % train data | 4% | 6% | 8% | 10% | 20% | 40% | 60% | 80% |
|---|---|---|---|---|---|---|---|---|
| R20 | 51.5 | 60.2 | 68.6 | 68.8 | 78.0 | 82.4 | 85.6 | 87.4 |
| R20N (ours) | **65.9** | **69.2** | **74.1** | **74.2** | **81.7** | **85.8** | **88.6** | **89.3** |
| Gain | +14.4 | +9.0 | +5.5 | +5.4 | +3.7 | +3.4 | +3.0 | +1.9 |

Table A4: **Downstream classification on ViTs.** We evaluate the performance of a small ViT [39] on CIFAR-10 on our proposed benchmarks using the raw image as input (left column) and feature maps from ResNets. ViT exhibits more robustness than vanilla ResNets (Table A1), but using feature maps from our normalized convolutions significantly reduces the performance drop after perturbations compared to all the other baselines.

|  | ViT [39] input type | | | | | | |
|---|---|---|---|---|---|---|---|
|  | Image $(I)$ $\in \mathbb{R}^{32\times32}$ | ResNet feature maps $(f \in \mathbb{R}^{16\times16})$ | | | | | |
|  |  | $R20_f$ | $R20N_f$ | $R32_f$ | $R32N_f$ | $R44_f$ | $R44N_f$ |
| $D_C$ | 85.4 | 86.4 | **88.8** | 84.7 | **88.5** | 85.9 | **89.4** |
| $D_L$ | 83.1 | 82.4 | **87.2** | 80.4 | **87.6** | 81.1 | **88.0** |
| $D_B$ | 80.2 | 79.2 | **80.2** | 77.6 | **81.9** | 79.2 | **82.6** |
| $D_S$ | 80.6 | 39.9 | **84.4** | 32.6 | **84.7** | 28.7 | **84.4** |

and without any normalization layer ($\times$), where we get the best results on most benchmarks while maintaining original test set accuracy. See § 4.3 for more details.

Table A5: **Ablation with normalization layers.** We train ResNets on CIFAR-10 with different normalization layers using vanilla convolutions and our normalized convolutions. We get best results on most benchmarks while maintaining original test set accuracy.

|  | ResNet20 | | | ResNet20N (ours) | | | ResNet32 | | | ResNet32N (ours) | | | ResNet44 | | | ResNet44N (ours) | | |
|---|---|---|---|---|---|---|---|---|---|---|---|---|---|---|---|---|---|---|
|  | $\times$ | IN | BN | $\times$ | IN | BN | $\times$ | IN | BN | $\times$ | IN | BN | $\times$ | IN | BN | $\times$ | IN | BN |
| $D$ | 89.0 | 89.4 | 91.4 | 88.1 | 90.1 | **91.5** | 88.8 | 90.0 | **92.0** | 89.6 | 90.7 | **92.0** | 89.2 | 90.5 | **93.2** | 90.4 | 90.7 | **93.2** |
| $D_C$ | 86.2 | 88.8 | 87.9 | 87.5 | 89.7 | **91.1** | 85.8 | 90.0 | 87.7 | 88.9 | 90.2 | **91.5** | 86.6 | 90.3 | 89.7 | 89.5 | 90.4 | **92.6** |
| $D_L$ | 83.8 | 87.5 | 84.2 | 86.3 | 88.8 | **89.6** | 82.7 | 88.4 | 84.3 | 87.9 | 89.1 | **90.4** | 82.7 | 89.2 | 86.4 | 88.3 | 89.9 | **92.0** |
| $D_B$ | 80.7 | 87.0 | 82.9 | 81.6 | **87.6** | 85.5 | 80.8 | 88.2 | 80.1 | 83.1 | 88.1 | **87.6** | 81.5 | 88.8 | 87.3 | 83.6 | 88.5 | **87.8** |
| $D_S$ | 43.5 | 84.0 | 38.1 | 85.9 | 79.4 | **89.5** | 42.9 | 88.1 | 28.6 | 85.9 | 79.2 | **88.8** | 43.2 | 89.3 | 49.5 | 86.9 | 85.4 | **90.2** |

## A.5 Normalization as soft regularization

We compare our approach to weight normalization through soft regularization. Specifically, given the set of convolutional filter weights $W$ in a CNN, where each filter weight is a 3D tensor $w \in \mathbb{R}^{C\times M\times N}$, the regularization term is given by

$$R(W) = \sum_{w\in W} \left( \left|1 - \|w^+\|\right| + \left|1 - \|w^-\|\right| \right). \tag{14}$$

The total loss function with regularization can be written as $L_{total} = L + \alpha R(W)$, where $\alpha = 0.01$ controls the strength of the regularization. We add this regularization to the loss during training of a R20 on CIFAR-10 using the same hyperparameters from § 4.1. In Table A6 we show that regularization improves robustness over vanilla training (second row), but our approach yields significantly better results (bottom row). While regularization may reduce $|r(w)|$ it does not guarantee $|r(w)| = 0$ for all filters like our approach (Fig. 13).

## A.6 Non-linearity impacts

Our proposed method ensures atmosphere-equivariance at the convolutional layer level. That is, prior to any non-linear activation, filter responses transform in a structured and predictable way under global intensity shifts (gain and bias). This forms the foundation for building robustness to such transformations throughout the network.

Table A6: **Our normalized convolutions beat normalization as soft regularization.** Adding the soft regularization (✓) improves the robustness of vanilla R20 (see first two rows), but our approach surpasses soft regularization (see the two bottom rows). Underlined values show the best accuracy against their non-regularized counterpart, while **bold** values show the best overall.

| Model | soft reg. | $D$ | $D_C$ | $D_L$ | $D_B$ | $D_S$ |
|-------|-----------|------|-------|-------|-------|-------|
| R20   |           | 91.4 | 87.9  | 84.2  | 82.9  | 38.1  |
| R20   | ✓         | **92.0** | 88.8 | 85.9 | 84.1 | 48.8 |
| R20N  |           | 91.5 | 91.1  | 89.6  | 85.5  | **89.5** |
| R20N  | ✓         | **92.0** | **91.5** | **90.5** | **86.0** | 89.0  |

It is important to clarify that, while individual layers are equivariant, the goal of the full network is to achieve invariance to atmospheric transformations—just as classical CNNs preserve spatial equivariance locally but aim for global translational invariance in their final semantic outputs. Non-linearities such as ReLU serve this purpose: they progressively reduce sensitivity to irrelevant variations (e.g., illumination), transforming equivariant features into invariant representations through depth and learned hierarchy.

What makes our approach effective is that starting from an atmosphere-equivariant basis makes the job of learning invariance easier and more structured. This is evident in our experiments (Table 3), where models using filter normalization without any data augmentation outperform baselines with augmentation, by a large margin (e.g., +14% accuracy on $D_S$). This suggests that the network is better able to learn invariances downstream when the early layers encode meaningful, interpretable transformations.

In short, while non-linearities do break strict equivariance, they play a necessary and constructive role in turning structured responses into invariant representations. Our method complements this process by providing a more robust and interpretable foundation at the convolutional level.

We did further experiments on different types of activation functions, specifically Tanh and Sigmoid, in place of ReLU. Our results on CIFAR-10 $D_C$ with R20 and our R20N model show that our method consistently outperforms the vanilla R20 model, regardless of the activation function used (see Table A7).

Table A7: Filter normalization provides robustness benefits that are complementary to the choice of activation function, as demonstrated by experiments.

| Model | Tanh | Sigmoid |
|-------|-------|---------|
| R34   | 90.5% | 88.1%   |
| R34N  | **91.5%** | **91.5%** |

This suggests that our filter normalization approach provides robustness benefits that are complementary to the choice of activation function.

## A.7    Further experimental validation

Our method is specifically designed to address atmospheric transformations—i.e., global additive and multiplicative corruptions such as changes in brightness, contrast, and haze (see Eq. 10 and Fig. 2). We already evaluate our method across a range of corruption types and intensities (see also Fig. A.1.1).

ImageNet-C [14] includes severe and often localized distortions (e.g., noise, blur, digital artifacts) that fall outside the atmospheric transformation family we explicitly target. Our method outperforms on some categories such as brightness and underperforms on others such as snow, all with a very small margin. While we acknowledge these results, we emphasize that ImageNet-P, proposed in the same paper, offers a more direct test of our model's design goals: ImageNet-P applies smooth, progressive perturbations, especially in the "weather" category (e.g., brightness and snow), aligning well with our focus on global intensity transformations.

Robustness is measured using the Flip Rate (FR)—the fraction of samples whose predicted class changes at the highest perturbation level compared to the clean version: $FR = \frac{1}{n}\sum_{i=1}^{n} \mathbb{1}(f(x_i) \neq$

$f(x_i^{30})$), where $x_i^{30}$ is the corrupted image at the maximum perturbation level (30), and $n$ are all images included in the "weather" category, which includes brightness and snow modifications (See Table A8).

The flip rate of a vanilla R34 is 41.3%, while our R34N with normalized convolutions achieves 40.0% flip rate (lower is better).

Table A8: **Results on ImageNet-P.** We show the Top-1 accuracy (↑) and the Flip Rate (FR ↓) for ImageNet-P, focusing on weather corruptions (i.e., brightness and snow). The flip rate of a vanilla R34 is 41.3%, while our R34N with normalized convolutions achieves 40.0% flip rate (lower is better). While both metrics show consistent trends, the FR gap is more pronounced than the accuracy gap, making it a more sensitive indicator of perturbation robustness.

| Model | Accuracy (↑) | Flip Rate (FR ↓) |
|---|---|---|
| R34 | 44.7 | 41.3 |
| R34N | **44.9** | **40.0** |

# B    R34 training curves on ImageNet

Here we show the training curves of the vanilla R34 and our R34N on ImageNet-1k. We train both networks from scratch on the original ImageNet-1k training set for 90 epochs, a batch size of 256, and SGD with an initial learning rate of 0.1 divided by 10 every 30 epochs. In Fig. A2 we can see that our R34N exhibits a more stable validation accuracy throughout training.

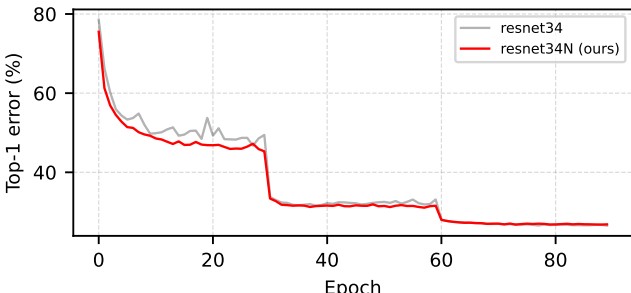

Figure A2: **Filter normalization regularizes learning showing more stable validation accuracy during training.** ResNet34 training curve on ImageNet-1k with the hyperparameters described in the ImageNet experiments in § 4.1.

# C    Feature visualizations

In Fig. A3 we show the learned filters from the first layer of R34 (left) and our R34N (right) models, trained on ImageNet-1k using the hyperparameters in § 4.1.Our model with normalized convolutions (R34N) exhibits more diverse features, consistent with the results in Fig. 12.

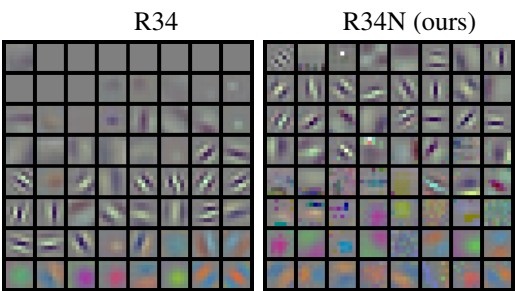

Figure A3: **Filter visualizations.** Learned filters from the first layer of R34 (left) and R34N (right) models, trained on ImageNet-1k. Our R34N exhibits more diverse features.

