# OpenReview forum: "Normalize Filters! Classical Wisdom for Deep Vision"
_NeurIPS.cc/2025/Conference — NeurIPS 2025 poster_

### Official Review · Reviewer_PcoK · 2025-06-29

**Clarity:** 3
**Significance:** 3
**Originality:** 3
**Rating:** 5
**Confidence:** 4

**Summary:**

This paper uncovers that filter kernels in convolutional neural networks (CNNs) are usually not normalized (meaning that the sum of all positive respectively all negative components is 0 or 1 respectively 0 or -1). Classic results from image processing show that such filters are not equivariant to gain and bias changes to their input, which could hinder generalization to gains and biases which were not seen during training. A simple formulation to normalize the filters in the abovementioned way is proposed, and the subsequent equivariance to gain and bias (and even invariance to bias in the usual case of filters with both positive and negative components) is proven. Very extensive evaluation demonstrates that these properties actually improve image classification accuracy under unseen gains and biases in many cases, and various analyses show that this method makes networks substantially more well-behaved.

**Questions:**

1. Looking at Eq. 1, I wonder whether and how these results also apply to fully connected layers?
2. Line 174f: To a cursory reader, it might be non-obvious that after the normalization, is is guaranteed that w.l.o.g., $||w^+|| = 1$ and $||w^-|| =$ either 1 or 0, and hence $r(w)$ is either 0 or 1, thus the filter is either strictly averaging or strictly differencing, and Eq. 11 can be used to prove the equivariance and invariance guarantees. I suggest clearly describing this reasoning and really "hammering it in", as I believe it is a key step to understanding your contribution.
3. If I'm not mistaken, the theoretical equivariance and invariance results only cover spatially uniform gain and bias. As you're also evaluating on spatially varying gain and bias, is it possible to elaborate on that? Do you just assume local spatial uniformity since filter kernels are usually small? If yes, I think this should be mentioned in the paper.
4. What is meant by "co-domain symmetry", mentioned in various places in the paper?
5. Line 212f: Is there a reason why you introduce sort pooling layers? In general, I would appreciate a more complete description of all tested architectures, including network width, depth, activation functions, pooling layers, residual connections, and the use of batch normalization in the interest of reproducibility.
6. Lines 214 & 243: Is there a specific reason why you chose SGD over Adam?
7. Lines 326f: "Artifacts reduction" and "a more targeted correction" are rather handwavy justifications for the superiority of filter normlization over instance normalization. In addition, no intuition to what this actually means is provided. I suggest you either elaborate on this or remove these claims.

Furthermore, here are some minor questions and potential improvements:
- Lines 191 & 202: I don't understand what the $\mathbf{1}()$ function in the definitions of $\tilde{x}$ is doing.
- Lines 194 & 198: Shouldn't it be the Hadamard product $\circ$ in the definitions of $\tilde{x}$ here?
- Fig. 5: Please add small labels to the vertical axes of the plots so one can grasp how large the accuracy improvement really is.
- Fig. 6: Could you specify in the paper what "moderate" and "strong" corruption are concretely?
- Fig. 9: Similarly, could you specify which corruption you used here?

**Ethical Concerns:**

["NO or VERY MINOR ethics concerns only"]

**Final Justification:**

The authors promise to fix various ambiguous or unclear phrasings both the other reviewers and I stumbled over, further improving their already well-written manuscript. All my technical concerns, as well as a major concern about the influence of nonlinearities brought up by reviewer t7kj, were addressed. Even though the evaluation is weak in some aspects (as pointed out by other reviewers) and the experiments show that this method mostly helps in extreme cases where data augmentation is insufficient (see limitations above), I think that alone constitutes a substantial and interesting enough contribution.

**Limitations:**

In many cases, it appears that image classification accuracy actually doesn't improve that much when introducing filter normalization. Big improvements are mostly seen when the bias is very high (evaluation set $D_S$), only few images are used for training (Fig. 7), or the ExDark dataset is used.

**Paper Formatting Concerns:**

I'm not fully sure whether this is a formal problem, but some figure captions seem to be awfully close to the main text (e.g., Fig. 13 & 14). I think a little bit more breathing room would help with legibility.

**Quality:**

3

**Strengths And Weaknesses:**

Strengths:
1. The paper is well-written, introduces the problems with non-normalized filters very clearly, relates them nicely to classic results from image processing, and formally proves its claims.
2. The proposed change to CNNs is very non-invasive, elegant, and simple to implement. It could thus enjoy widespread adoption.
3. The evaluation is very extensive and doesn't hide the fact that in some cases, this method doesn't actually improve performance, but at least retains it. In addition, various behavioral analyses of the networks provide interesting insights, e.g., on hidden feature distribution or saliency.

Weaknesses:
1. I think section 3 could be streamlined a bit more to make it even easier to digest and follow. For some ideas, see my questions below.
2. The mix of evaluation plots and tables, also including the supplementary material, seems a bit messy.

---

> ### Author Rebuttal · Authors · 2025-07-30
>
> Dear reviewer PcoK,
>
> Thank you for your valuable feedback and comments. We appreciate your recognition of our paper's clear presentation of the problems with non-normalized filters, its connection to classic results from image processing, and the formal proof of our claims. We are also grateful for your praise of our proposed method's simplicity, elegance, and potential for widespread adoption. Additionally, we appreciate your acknowledgment of the thoroughness of our evaluation, including the insightful behavioral analyses that provide a deeper understanding of the networks. We address your concerns and questions in the response below.
>
> > I think section 3 could be streamlined a bit more to make it even easier to digest and follow. For some ideas, see my questions below.
>
> We appreciate the reviewer's suggestions below for streamlining section 3 and will incorporate them to improve clarity and readability.
>
> > The mix of evaluation plots and tables, also including the supplementary material, seems a bit messy.
>
> Thank you for your feedback about this. We agree that the current presentation of evaluation plots and tables, including the supplementary material, could be improved. With a large number of results to report, we strive to strike a balance between clarity and concision. In the final version, we will make every effort to polish the presentation, potentially moving some results to the appendix to enhance the overall flow and readability.
>
> **Questions:**
> > Looking at Eq. 1, I wonder whether and how these results also apply to fully connected layers?
>
> Great suggestion. Since fully connected (FC) layers can be viewed as a special case of convolutional layers, we ran a simple experiment to explore the effects of normalizing the linear layer in the classifier of an R20 trained on CIFAR-10, where our initial findings show a slight increase in accuracy. Interestingly, this raises connections to classical wisdom on classifier design, such as maximizing margin with weight regularization, similar to SVMs. This potential link highlights opportunities for future work, where we can further analyze and explore the relationships between normalization and classifier performance.
>
> > Line 174f: To a cursory reader, it might be non-obvious that after the normalization, is s guaranteed that w.l.o.g.,  and  either 1 or 0, and hence  is either 0 or 1, thus the filter is either strictly averaging or strictly differencing, and Eq. 11 can be used to prove the equivariance and invariance guarantees. I suggest clearly describing this reasoning and really "hammering it in", as I believe it is a key step to understanding your contribution.
>
> We will follow the reviewer’s suggestion and describe this part more clearly to facilitate the understanding of our contribution.
>
> > If I'm not mistaken, the theoretical equivariance and invariance results only cover spatially uniform gain and bias. As you're also evaluating on spatially varying gain and bias, is it possible to elaborate on that? Do you just assume local spatial uniformity since filter kernels are usually small? If yes, I think this should be mentioned in the paper.
>
> Thank you for pointing this out! We indeed assumed local spatial uniformity, which holds reasonably well given the small size of filter kernels. You are correct that this assumption might not strictly hold for spatially varying gain and bias. Notably, our experiments show that the gain in accuracy from our method compared to vanilla ResNets is more pronounced for spatially varying cases. For instance, our R34N obtains a 4.2% gain in accuracy on $D_C$, whereas the gains on $D_L$ and $D_B$   are 7.2% and 31.3%, respectively. This suggests that our method provides more significant benefits when dealing with spatially varying corruptions, particularly those with faster spatial variations like $D_B$. We'll make sure to clarify this assumption and its implications in the paper to provide further context.
>
> > What is meant by "co-domain symmetry", mentioned in various places in the paper?
>
> In classical signal processing, spatial equivariance is well understood: A convolutional filter responds consistently to translations of the input. Our filter normalization extends this structured response into the co-domain, i.e., the filter output space under intensity changes.
>
> We define co-domain symmetry as follows: A filter exhibits co-domain symmetry if its response transforms in a predictable, structured way under affine transformations of the input intensity $x$, via gain ($g$) and offset ($o$).
>
> We will clarify these definitions in the revision and illustrate them more explicitly in the discussion around Eq. (11).
>
> > Line 212f: Is there a reason why you introduce sort pooling layers? In general, I would appreciate a more complete description of all tested architectures, including network width, depth, activation functions, pooling layers, residual connections, and the use of batch normalization in the interest of reproducibility.
>
> We added affine convolutions and sort pooling layers for a fairer comparison with norm-equivariant nets [15]. We will add a section in the appendix giving more details about the architecture used for this experiment.
>
> [15] Herbreteau, et al. Normalization-equivariant neural networks with application to image denoising. NeurIPS, 2023.
>
> > Lines 214 & 243: Is there a specific reason why you chose SGD over Adam?
>
> Not really. We followed implementations from PyTorch official examples.
>
> > Lines 326f: "Artifacts reduction" and "a more targeted correction" are rather handwavy justifications for the superiority of filter normlization over instance normalization. In addition, no intuition to what this actually means is provided. I suggest you either elaborate on this or remove these claims.
>
> Regarding "artifacts reduction", we observed that instance normalization can sometimes introduce undesirable artifacts in the output, particularly when the instance statistics are not representative of the atmospheric corruptions. On the other hand, by "more targeted correction", we mean that normalizing the weights rather than activations allows our approach to directly correct the filter's response to atmospheric corruptions, rather than indirectly adjusting the activations. We will further analyze our claims and expand on these points in the final version of the paper.
>
> **Minor comments:** We will carefully consider all minor comments and revise the paper accordingly for the final version.

---

> > ### Comment · Reviewer_PcoK · 2025-08-04
> >
> > Thanks for the detailed responses. Everything I was unsure about was cleared up, and I'm glad you will revise the paper so that other readers don't stumble across the same questions. Additionally, I really enjoyed your response on the influence of nonlinearities brought up by reviewer t7kj, and am happy that you will include those insights in the final revision. I'll keep my score as-is.

---

> > > ### Author Response · Authors · 2025-08-06
> > >
> > > We are pleased to hear that your concerns have been addressed, and we sincerely appreciate your recognition of our contributions.

---

### Official Review · Reviewer_qNTu · 2025-07-02

**Clarity:** 3
**Significance:** 3
**Originality:** 3
**Rating:** 4
**Confidence:** 5

**Summary:**

This paper introduces a new filter normalization technique to enhance the capability of image processing of deep convolutional networks. Classical / human-crafted image filters are usually normalized to maintain consistency, interpretability, and to avoid visual artifacts. Well the data-driven image filters usually do not have this feature. However, the lack of normalization would lead to distorted filter responses, especially under atmospheric / lighting change environment. The proposed method discovers that all the convolutional filters can be reformulated as a combination of averaging filter and differencing filter, and hence proposes to normalize any given / learnt convolutional filter to ensure the consistency and interpretability. Extensive experiments on a wide range of application show the effectiveness of the proposed filter normalization method.

**Questions:**

This paper is technically sound and well formulated. The authors are suggested to address the weakness in the weakness session during rebuttal period.

**Ethical Concerns:**

["NO or VERY MINOR ethics concerns only"]

**Final Justification:**

I would like to keep my original ratings for this submission as the authors have adopted the comments and addressed the weakness.

**Limitations:**

yes

**Paper Formatting Concerns:**

No concerns on formatting

**Quality:**

3

**Strengths And Weaknesses:**

### Strength

1. The proposed method comes with a clear and elegant formulation and solid derivation and proof.
2. The observation of the unnormalized convolutional filter in deep learning is critical and useful. This fact could potentially discover some intrinsic features of convolutional deep neural networks.
3. The paper is strongly motivated and clearly structured. The proposed method could be applied to many real application problems such as photo editing, intensity correction, astronomic image processing, etc.
4. The extensive experiments show that the proposed method has achieved the beset performance across different application scenarios.


### Weakness

1. The header of section 3.1 "Any filter is both averaging and differencing" is not guaranteed. It should be better rephrased as "any linear / convolutional filter ... "
2.  The experiment on data augmentation on CIFAR-10 with only atmospheric augmentations may not be very representative.  First, the data size of CIFAR and the image size are both small. Recent advances in image processing are all conducted on large scale high resolution data. It is suggested to make more thorough experiments on those dataset to make augmentation conclusion. Second, the experiment results do not show a clear trend that data augmentation techniques make model suffer, even though the performance on D_s alone is not as good as expected.

---

> ### Author Rebuttal · Authors · 2025-07-30
>
> Dear reviewer qNTu,
>
> Thank you for your valuable feedback and comments. We appreciate your recognition of our method's clear and elegant formulation, solid derivation, strong motivation, well-structured presentation, and comprehensive experiments. We also appreciate your acknowledgment of the importance and usefulness of our observation regarding unnormalized filters in deep networks. Below, we address the concerns and questions you raised.
>
> > The header of section 3.1 "Any filter is both averaging and differencing" is not guaranteed. It should be better rephrased as "any linear / convolutional filter ... "
>
> Thank you for the suggestion. We completely agree that specifying 'linear filter' adds precision and will improve clarity to the title of section 3.1. We will adopt this rephrased title in the final version.
>
> > The experiment on data augmentation on CIFAR-10 with only atmospheric augmentations may not be very representative. First, the data size of CIFAR and the image size are both small. Recent advances in image processing are all conducted on large scale high resolution data. It is suggested to make more thorough experiments on those dataset to make augmentation conclusion. Second, the experiment results do not show a clear trend that data augmentation techniques make model suffer, even though the performance on D_s alone is not as good as expected.
>
> Thank you for your insightful comment. While our results on CIFAR-10 don't show a clear trend of data augmentation techniques hurting model performance, our approach achieves similar robustness without the need for data augmentations. This demonstrates a key advantage of our method, as it eliminates the need for data augmentation and the associated computational costs. Additionally, our approach avoids the challenges of designing effective augmentations, which can require domain-specific knowledge. To further validate our findings, we will include experiments on ImageNet in the final version, when computational resources permit.

---

### Official Review · Reviewer_j6NF · 2025-07-03

**Clarity:** 4
**Significance:** 3
**Originality:** 3
**Rating:** 5
**Confidence:** 3

**Summary:**

The paper proposes to separate and normalize filters in convolutional layers into positive and negative filters to help mitigating against gain and bias. They show experiments with a neural networks modified this way and compare with other computer vision models .

**Questions:**

1. Would it be possible to also add a fine-tuned CLIP for comparison
2. Would it be possible to add a non-customized dataset to actually see how the modification to the filters behave when corruptions are not just gain and bias?

**Ethical Concerns:**

["NO or VERY MINOR ethics concerns only"]

**Final Justification:**

Authors also addressed enough of issues, and my score was already at 5 which I will keep.

**Limitations:**

yes

**Quality:**

3

**Strengths And Weaknesses:**

**Strengths**
* Simple modification to convolutional layers with apparent gains in robustness and interpretation

**Weaknesses**
* experiment could include other corruptions beyond these customized datasets.
* comparison of ResNet fine-tuned with CLIP on zero-shot is somewhat misleading, and perhaps not the best choice. We know that CLIP does not do well on robustness to covariate shift, I am not sure what the experiment shows for the actual topic of the paper.
* not widely accepted use of the word "atmosphere"  effect, The paper even shows some experiments on astronomy data taken from space.

---

> ### Author Rebuttal · Authors · 2025-07-30
>
> Dear reviewer j6NF,
>
> Thank you for your valuable feedback and comments. We appreciate your recognition of our method’s simplicity and the gains it provides in robustness and interpretation. We address your concerns and questions in the response below.
>
> > Experiment could include other corruptions beyond these customized datasets.
>
> Thank you for the suggestion to strengthen our experimental validation using established robustness benchmarks. Our method is specifically designed to address atmospheric transformations—i.e., global additive and multiplicative corruptions such as changes in brightness, contrast, and haze (see Eq. 10 and Fig. 2). We already evaluate our method across a range of corruption types and intensities (see also Fig. A.1.1 in the Appendix).
>
> ImageNet-C includes severe and often localized distortions (e.g., noise, blur, digital artifacts) that fall outside the atmospheric transformation family we explicitly target.  Our method outperforms on some categories such as brightness and underperforms on others such as snow, all with a very small margin.  While we acknowledge these results, we emphasize that ImageNet-P, proposed in the same paper, offers a more direct test of our model’s design goals: ImageNet-P applies smooth, progressive perturbations, especially in the "weather" category (e.g., brightness and snow), aligning well with our focus on global intensity transformations.
>
> Robustness is measured using the Flip Rate (FR)—the fraction of samples whose predicted class changes at the highest perturbation level compared to the clean version:  $FR = \frac{1}{n}\sum_{i=1}^n \mathbb{1}(f(x_i) \neq f(x_i^{30}))$, where $x_i^{30}$ is the corrupted image at the maximum perturbation level (30), and $n$ are all images included in the ”weather” category, which includes brightness and snow modifications (See results below).
>
> | Model | FR weather ($\downarrow$) |
> | ----------- | ----------- |
> | R34 | 41.3% |
> | R34N (ours) | **40.0%** |
>
> The flip rate of a vanilla R34 is 41.3%, while our R34N with normalized convolutions achieves 40.0% flip rate (lower is better).
>
> Lastly, we further assess generalization to natural variation by binning the ImageNet-1k test set into 9 contrast levels (based on pixel standard deviation). R34N maintains more stable accuracy across the contrast spectrum compared to the baseline, which drops sharply at extreme values (see Fig. X).  This confirms the model's improved robustness to natural realistic atmosphere conditions.
>
> | Model | Lowest | Contrast 2 | Contrast 3 | Contrast 4 | Contrast 5 | Contrast 6 | Contrast 7 | Contrast 8 | Highest|
> | ----------- | ----------- | ----------- | ----------- | ----------- | ----------- | ----------- | ----------- | ----------- | ----------- |
> |R34 	| 65.4% |  74.0% | 74.8% | 75.2% | 73.2% | 70.8% | 70.7% | 67.9% | 63.3%|
> |R34N 	| **69.2%** | 74.1% | 74.6% | 74.5% | 73.3% | 71.1% | 71.1% | 67.3% | **71.4%**|
>
> We will include new plots, statistical analyses, and representative image examples in the final version of the paper.
>
> [1] Hendrycks, et al. Benchmarking Neural Network Robustness to Common Corruptions and Perturbations. ICLR 2019.
>
> > Comparison of ResNet fine-tuned with CLIP on zero-shot is somewhat misleading, and perhaps not the best choice. We know that CLIP does not do well on robustness to covariate shift, I am not sure what the experiment shows for the actual topic of the paper.
>
> We appreciate the reviewer’s point and fully acknowledge that comparing our R34N model—trained on ImageNet-1k for supervised classification—to CLIP, a foundation model trained on the much larger LAION-5B dataset using contrastive vision-language pretraining, involves substantial differences in scale, objective, and intended use.
>
> That said, if the comparison is “unfair,” we would argue it is unfair to our disadvantage: CLIP is trained on several orders of magnitude more data with broader supervision, yet still shows notable vulnerability to simple intensity corruptions, where our much smaller model remains robust. From this perspective, the comparison is not only informative but also conservative, demonstrating that large-scale pretraining does not guarantee robustness to common low-level variations.
>
> Our intent is not to claim general superiority over CLIP, but to highlight complementary strengths: CLIP excels in semantic generalization and zero-shot transfer, while our method targets physical robustness to real-world image degradations. We believe this suggests that classical filtering principles—like those embedded in our filter normalization—may provide value even when incorporated into large-scale models, particularly CNN-based ones such as CvT [2] and ConvNeXt [3].
>
> We will revise the manuscript to more clearly acknowledge the training scale differences and provide a balanced discussion of this comparison.
>
> [2] Wu, et al. CvT: Introducing Convolutions to Vision Transformers. ICCV 2021.
>
> [3] Liu, et al. A ConvNet for the 2020s. CVPR 2022.
>
> > Not widely accepted use of the word "atmosphere" effect, The paper even shows some experiments on astronomy data taken from space.
>
> We borrowed the term 'atmosphere' from Adelson’s work on Lightness Perception and Lightness Illusions [4], where it is defined as 'The net effect of the viewing conditions, including additive and multiplicative effects, may be termed an “atmosphere.”' We find this term suitable for our purposes. However, as the reviewer pointed out, we agree it may cause confusion when applied to domain-specific data, such as astronomy images. To clarify, we will specify in the paper that the term 'atmospheric effects' is particularly relevant to natural color images.
>
> [4] Adelson. Lightness Perception and Lightness Illusions. MIT Press, 2000.
>
> **Questions:**
> > Would it be possible to also add a fine-tuned CLIP for comparison
>
> Unfortunately, fine-tuning CLIP is not feasible for the length of the discussion period. We included zero-shot CLIP as a baseline since it is a widely used approach for image classification.

---

> ### Author Response · Authors · 2025-08-07
>
> Thanks much for your careful and thoughtful reviews.
>
> We would like to clarify that our comparison between R34N and CLIP focuses on robustness to atmospheric corruptions, not semantic generality.
>
> In the table below, we can see that specialist models like ResNets and ViTs trained on ImageNet for supervised classification (first three rows) perform well on the ImageNet test set ($D$) but struggle with corrupted datasets (for instance, see ~1-2% accuracies on $D_S$). CLIP, a generalist model trained on 400M images, performs better on corrupted datasets due to its exposure to more variations (See accuracy of up to 49% on $D_S$). However, our model, R34N, outperforms CLIP in robustness to atmospheric corruptions (67% on $D_S$), despite not seeing any such transformations during training.
>
> | Model | # Params. | $D$ | $D_C$ | $D_L$ | $D_B$ | $D_S$ |
> |---|---|---|---|---|---|---|
> | R34 | 22 | 73.3 | 68.6 | 64.5 | 36.8 | 2.1 |
> | R101 | 45 | 77.4 | 69.1 | 62.9 | 34.0 | 1.1 |
> | ViT-L | 304 |  **79.3**  |  **72.8**  | 67.7 | 41.8 | 1.1 |
> | CLIP-B/32 | 151 | 60.2 | 58.4 | 56.4 | 52.3 | 14.3 |
> | CLIP-L/14 | 428 | 68.1 | 67.4 | 66.7 | 65.6 | 49.1 |
> | R34N (ours) | 22 | 73.2 |  **72.8**  |  **71.7**  |  **68.1**  |  **67.0**  |
>
> Despite CLIP’s broader training, it suffers a significant drop under intensity corruptions, while our much smaller and more narrowly trained model remains robust—even to perturbations it never saw.  From this perspective, the comparison is informative: it highlights that large-scale pretraining does not automatically confer robustness to low-level corruptions, and that classical principles like filter normalization can still play a valuable role.
>
> We consider our comparison with CLIP is not unfair as our goal is not to claim general superiority over CLIP, but to illustrate an aspect where even foundation models may benefit from architectural improvements. Filtering-based normalization remains relevant, particularly for CNN-based models such as CvT [1] and ConvNeXt [2].
>
> We will revise the paper to reflect this nuanced comparison and clarify that our claims are specific to robustness.
>
> [1] Wu, et al. CvT: Introducing Convolutions to Vision Transformers. ICCV 2021.
>
> [2] Liu, et al. A ConvNet for the 2020s. CVPR 2022.

---

### Official Review · Reviewer_t7kj · 2025-07-05

**Clarity:** 2
**Significance:** 4
**Originality:** 3
**Rating:** 5
**Confidence:** 3

**Summary:**

The paper addresses the issue of deep learning models' reduced robustness to intensity variations in images, which stems from the unnormalized nature of learned convolutional filters. It proposes a novel Filter Normalization method that decomposes convolutional filters into their averaging filter and differencing filter components and normalizes them individually. This modification is designed to ensure that filters exhibit atmosphere-equivariance, leading to consistent and interpretable responses under varying illumination conditions. Experiments on both artificially corrupted and natural datasets demonstrate that this approach significantly enhances model robustness and generalization across various CNN and Vision Transformer architectures. The authors claim that this method, by integrating classical filtering principles, enables leaner models to outperform larger, unnormalized counterparts, including CLIP, on these specific benchmarks.

**Questions:**

- __Clarify Theoretical Concepts:__ Provide clear mathematical definitions for "co-domain symmetry" and "atmosphere-equivariance." Elaborate on how Equation (11) directly translates to the broader concept of atmosphere-equivariance for deep networks.

- __Address Non-linearity Impacts:__ Discuss the implications of non-linear activation functions (e.g., ReLU) on the proposed filter normalization's effects and the concept of "interpretable responses" throughout the entire network.

- __Refine CLIP Comparison:__ Rephrase claims regarding CLIP's "outperformance" to explicitly acknowledge and emphasize the vast disparity in training data scale (ImageNet-1k vs. LAION-5B).

- __Strengthen Experimental Validation:__ Evaluate the proposed method on well-established public robustness benchmarks, particularly relevant subsets of ImageNet-C (e.g., brightness, contrast, haze). Expand the natural data experiments beyond just inline text, presenting them in comprehensive tables that compare all baselines where applicable.

**Ethical Concerns:**

["NO or VERY MINOR ethics concerns only"]

**Final Justification:**

The authors' responses adequately address the theoretical and non-linearity concerns, and their re-framing of the CLIP comparison is reasonable. I strongly recommend that the authors change the claim on the CLIP comparison.
Moreover, the authors adequately added some important additional experiments.

For these reasons, I increase my rating by 1.
Thank you.

**Limitations:**

yes

**Paper Formatting Concerns:**

There's no formatting issue.

**Quality:**

3

**Strengths And Weaknesses:**

# Strengths

### 1. Addresses a Critical Problem in Deep Learning
The paper tackles the significant and long-standing issue of robustness in neural networks, particularly concerning image intensity variations, which is crucial for real-world applications like autonomous driving and medical imaging.

### 2. Novelty and Effectiveness of Filter Normalization
The core idea of normalizing convolutional filters by separating positive and negative parts and applying learnable scaling and shifting is novel. This approach is well-motivated by classical filter theory. Its effectiveness is demonstrated by consistent performance gains across CNNs and Vision Transformers on both artificial and natural benchmarks.

### 3. Simplicity and Broad Applicability
The proposed method is simple to implement ("simple yet effective modification") and can be readily integrated into various convolution-based architectures, suggesting high potential for wide adoption and future development in different domains

# Weaknesses

### 1. Ambiguous Definition and Proof of Co-domain Symmetry and Atmosphere-Equivariance
The paper claims that its method ensures "atmosphere-equivariant, enabling co-domain symmetry", but it lacks clear mathematical definitions for these terms. While Equation (11) demonstrates linearity for individual filter responses, the logical connection between this and the general concept of "atmosphere-equivariance" or "co-domain symmetry" for the entire network remains insufficiently elaborated.

### 2. Insufficient Consideration of Non-linearities
The theoretical derivations for consistent responses (e.g., Equation 11) apply primarily to linear convolutional layers. The paper does not adequately discuss how the proposed filter normalization impacts or interacts with non-linear activation functions (like ReLU) within deep networks, which could dilute the "interpretable responses" and overall equivariance in a multi-layer network.

### 3. Unfair Comparison with CLIP
The paper strongly highlights outperforming CLIP on corrupted datasets. However, this comparison is unfair as CLIP (zero-shot) models are pre-trained on significantly larger datasets (e.g., LAION-5B) compared to the ImageNet-1k training of the proposed R34N. While the R34N demonstrates efficiency, presenting this as a direct outperformance of a model trained on orders of magnitude more data is misleading and should be heavily qualified. I'm not saying the necessity of a fair comparison, but the authors should not claim like "better than CLIP trained on much larger datasets".

### 4. Limited Scope and Public Credibility of Experimental Results
The majority of reported performance gains are on custom-designed, artificially corrupted ImageNet and CIFAR-10 datasets. These simulated atmospheric effects might be overly simplistic and not fully representative of complex real-world conditions. While some results on natural datasets like ExDark and LEGUS are provided, they are limited (presented inline, not in comprehensive tables, and not compared against all baselines). Moreover, the absence of evaluations on well-established public benchmarks for robustness, such as relevant subsets of ImageNet-C (e.g., brightness, contrast, haze), diminishes the credibility and generalizability of the reported performance.

---

> ### Author Rebuttal · Authors · 2025-07-30
>
> Thank you for your valuable feedback and comments. We appreciate your acknowledgement that our work addresses a critical problem in Deep Learning, its novelty, effectiveness, simplicity, and broad applicability. We address your concerns and questions in the response below.
>
> > **Clarify Theoretical Concepts:** Provide clear mathematical definitions for "co-domain symmetry" and "atmosphere-equivariance." Elaborate on how Equation (11) directly translates to the broader concept of atmosphere-equivariance for deep networks.
>
> Thank you for raising the concern about the clarity of our definitions of co-domain symmetry and atmosphere-equivariance. We agree these terms need more precise explanation.
>
> In classical signal processing, spatial equivariance is well understood: A convolutional filter responds consistently to translations of the input. Our filter normalization extends this structured response into the co-domain, i.e., the filter output space under intensity changes.
>
> We define co-domain symmetry as follows: A filter exhibits co-domain symmetry if its response transforms in a predictable, structured way under affine transformations of the input intensity $x$, via gain ($g$) and offset ($o$).
>
> Our method ensures that normalized filters decompose into two interpretable components:
>
> 1. Averaging filters respond linearly to both gain and offset: $f(g \cdot x + o) = g \cdot f(x) + o$, where $|w| = 1$ and $r(w) = 1$.
>
> 2. Differencing filters are invariant to offset and equivariant to gain: $f(g \cdot x + o) = g \cdot f(x)$, where $|w| = 1$ and $r(w) = 0$.
>
> This decomposition yields atmosphere-equivariance, meaning that under global illumination changes (such as haze, shadows, or lighting shifts), the response of each filter changes in a predictable and interpretable way. For averaging filters, both contrast and brightness shift the output, while differencing filters respond only to contrast. This behavior is particularly desirable in scenarios where the semantic structure of the scene should be invariant to lighting conditions.
>
> We will clarify these definitions in the revision and illustrate them more explicitly in the discussion around Eq. (11).
>
> > **Address Non-linearity Impacts:** Discuss the implications of non-linear activation functions (e.g., ReLU) on the proposed filter normalization's effects and the concept of "interpretable responses" throughout the entire network
>
> Thank you for raising the question regarding the role of non-linearities (e.g., ReLU) and their impact on the proposed filter normalization and interpretability of responses.
>
> Our proposed method ensures atmosphere-equivariance at the convolutional layer level. That is, prior to any non-linear activation, filter responses transform in a structured and predictable way under global intensity shifts (gain and bias). This forms the foundation for building robustness to such transformations throughout the network.
>
> It is important to clarify that, while individual layers are equivariant, the goal of the full network is to achieve invariance to atmospheric transformations—just as classical CNNs preserve spatial equivariance locally but aim for global translational invariance in their final semantic outputs. Non-linearities such as ReLU serve this purpose: they progressively reduce sensitivity to irrelevant variations (e.g., illumination), transforming equivariant features into invariant representations through depth and learned hierarchy.
>
> What makes our approach effective is that starting from an atmosphere-equivariant basis makes the job of learning invariance easier and more structured. This is evident in our experiments (Lines 311–318 and Table 2), where models using filter normalization without any data augmentation outperform baselines with augmentation, by a large margin (e.g., +14% accuracy on $D_S$). This suggests that the network is better able to learn invariances downstream when the early layers encode meaningful, interpretable transformations.
>
> In short, while non-linearities do break strict equivariance, they play a necessary and constructive role in turning structured responses into invariant representations.  Our method complements this process by providing a more robust and interpretable foundation at the convolutional level.
>
> We did further experiments on different types of activation functions, specifically Tanh and Sigmoid, in place of ReLU. Our results on CIFAR-10 $D_C$ with R20 and our R20N model show that our method consistently outperforms the vanilla R20 model, regardless of the activation function used:
>
> | Model | Tanh | Sigmoid |
> | ----------- | ----------- | ----------- |
> | R34 | 90.5% | 88.1% |
> | R34N (ours) | **91.5%** | **91.5%** |
>
> This suggests that our filter normalization approach provides robustness benefits that are complementary to the choice of activation function.  We will include these experiments in the final version of our paper.
>
> > **Refine CLIP Comparison:** Rephrase claims regarding CLIP's "outperformance" to explicitly acknowledge and emphasize the vast disparity in training data scale (ImageNet-1k vs. LAION-5B).
>
> We appreciate the reviewer’s point and fully acknowledge that comparing our R34N model—trained on ImageNet-1k for supervised classification—to CLIP, a foundation model trained on the much larger LAION-5B dataset using contrastive vision-language pretraining, involves substantial differences in scale, objective, and intended use.
>
> That said, if the comparison is “unfair,” we would argue it is unfair to our disadvantage: CLIP is trained on several orders of magnitude more data with broader supervision, yet still shows notable vulnerability to simple intensity corruptions, where our much smaller model remains robust. From this perspective, the comparison is not only informative but also conservative, demonstrating that large-scale pretraining does not guarantee robustness to common low-level variations.
>
> Our intent is not to claim general superiority over CLIP, but to highlight complementary strengths: CLIP excels in semantic generalization and zero-shot transfer, while our method targets physical robustness to real-world image degradations. We believe this suggests that classical filtering principles—like those embedded in our filter normalization—may provide value even when incorporated into large-scale models, particularly CNN-based ones such as CvT [2] and ConvNeXt [3].
>
> We will revise the manuscript to more clearly acknowledge the training scale differences and provide a balanced discussion of this comparison.
>
> [2] Wu, et al. CvT: Introducing Convolutions to Vision Transformers. ICCV 2021.
>
> [3] Liu, et al. A ConvNet for the 2020s. CVPR 2022.
>
> > Strengthen Experimental Validation: Evaluate the proposed method on well-established public robustness benchmarks, particularly relevant subsets of ImageNet-C (e.g., brightness, contrast, haze). Expand the natural data experiments beyond just inline text, presenting them in comprehensive tables that compare all baselines where applicable.
>
> Thank you for the suggestion to strengthen our experimental validation using established robustness benchmarks. Our method is specifically designed to address atmospheric transformations—i.e., global additive and multiplicative corruptions such as changes in brightness, contrast, and haze (see Eq. 10 and Fig. 2). We already evaluate our method across a range of corruption types and intensities (see also Fig. A.1.1 in the Appendix).
>
> ImageNet-C includes severe and often localized distortions (e.g., noise, blur, digital artifacts) that fall outside the atmospheric transformation family we explicitly target.  Our method outperforms on some categories such as brightness and underperforms on others such as snow, all with a very small margin.  While we acknowledge these results, we emphasize that ImageNet-P, proposed in the same paper, offers a more direct test of our model’s design goals: ImageNet-P applies smooth, progressive perturbations, especially in the "weather" category (e.g., brightness and snow), aligning well with our focus on global intensity transformations.
>
> Robustness is measured using the Flip Rate (FR)—the fraction of samples whose predicted class changes at the highest perturbation level compared to the clean version:  $FR = \frac{1}{n}\sum_{i=1}^n \mathbb{1}(f(x_i) \neq f(x_i^{30}))$, where $x_i^{30}$ is the corrupted image at the maximum perturbation level (30), and $n$ are all images included in the ”weather” category, which includes brightness and snow modifications (See results below).
>
> | Model | FR weather ($\downarrow$) |
> | ----------- | ----------- |
> | R34 | 41.3% |
> | R34N (ours) | **40.0%** |
>
> The flip rate of a vanilla R34 is 41.3%, while our R34N with normalized convolutions achieves 40.0% flip rate (lower is better).
>
> Lastly, we further assess generalization to natural variation by binning the ImageNet-1k test set into 9 contrast levels (based on pixel standard deviation). R34N maintains more stable accuracy across the contrast spectrum compared to the baseline, which drops sharply at extreme values (see Fig. X).  This confirms the model's improved robustness to natural realistic atmosphere conditions.
>
> | Model | Lowest | Contrast 2 | Contrast 3 | Contrast 4 | Contrast 5 | Contrast 6 | Contrast 7 | Contrast 8 | Highest|
> | ----------- | ----------- | ----------- | ----------- | ----------- | ----------- | ----------- | ----------- | ----------- | ----------- |
> |R34 	| 65.4% |  74.0% | 74.8% | 75.2% | 73.2% | 70.8% | 70.7% | 67.9% | 63.3%|
> |R34N 	| **69.2%** | 74.1% | 74.6% | 74.5% | 73.3% | 71.1% | 71.1% | 67.3% | **71.4%**|
>
> We will include new plots, statistical analyses, and representative image examples in the final version of the paper.
>
> [1] Hendrycks, et al. Benchmarking Neural Network Robustness to Common Corruptions and Perturbations. ICLR 2019.

---

> ### Comment · Reviewer_t7kj · 2025-08-04
>
> The authors' responses adequately address the theoretical and non-linearity concerns, and their re-framing of the CLIP comparison is reasonable. I strongly recommend that the authors change the claim on the CLIP comparison.
>
> However, the experimental validation remains a significant concern. While the authors present promising Flip Rate (FR) results on ImageNet-P, the result does not include accuracy, which is a more fundamental metric.
>
> I understand that ImageNet-P is more appropriate than ImageNet-C, but is there any reason why the accuracy performance for ImageNet-P is not included in the rebuttal?

---

> ### Author Response · Authors · 2025-08-04
> **ImageNet-P accuracy results**
>
> Thank you for your request. We are happy to provide the accuracy results along with the Flip Rate (FR) below.  The accuracy of our R34N is higher than R34, which is consistent with its lower FR.
>
> | Model |  Accuracy ($\uparrow$) | Flip Rate ($\downarrow$) |
> | --- | --- | --- |
> | R34 | 44.7 | 41.3 |
> | R34N | **44.9**| **40.0** |
>
> We used FR as the metric in this experiment, following the convention established by the ImageNet-P benchmark [1] (Section 4.2). While both metrics show consistent trends, the FR gap is more pronounced than the accuracy gap, making it a more sensitive indicator of perturbation robustness — most likely the rationale behind its original proposal [1].  We are happy to include both metrics in the final version and hope this satisfactorily addresses your concern.
>
> [1] Hendrycks, et al. Benchmarking Neural Network Robustness to Common Corruptions and Perturbations. ICLR 2019

---

> > ### Comment · Reviewer_t7kj · 2025-08-06
> >
> > The authors' responses adequately address the theoretical and non-linearity concerns, and their re-framing of the CLIP comparison is reasonable. I strongly recommend that the authors change the claim on the CLIP comparison. Moreover, the authors adequately added some important additional experiments.
> >
> > For these reasons, I increase my rating by 1. Thank you.

---

> > > ### Author Response · Authors · 2025-08-06
> > >
> > > We are pleased to hear that your concerns have been addressed, and we sincerely appreciate your recognition of our contributions. Thank you for your thoughtful feedback and for increasing your rating.

---

> > > > ### Author Response · Authors · 2025-08-07
> > > >
> > > > As a final note, we appreciated the discussion and wanted to provide some final clarification regarding our comparison with CLIP. We understand that the fairness of this comparison has been raised, and we'd like to address this point.
> > > >
> > > > We would like to clarify that our comparison between R34N and CLIP focuses on robustness to atmospheric corruptions, not semantic generality.
> > > >
> > > > In the table below, we can see that specialist models like ResNets and ViTs trained on ImageNet for supervised classification (first three rows) perform well on the ImageNet test set ($D$) but struggle with corrupted datasets (for instance, see ~1-2% accuracies on $D_S$). CLIP, a generalist model trained on 400M images, performs better on corrupted datasets due to its exposure to more variations (See accuracy of up to 49% on $D_S$). However, our model, R34N, outperforms CLIP in robustness to atmospheric corruptions (67% on $D_S$), despite not seeing any such transformations during training.
> > > >
> > > > | Model | # Params. | $D$ | $D_C$ | $D_L$ | $D_B$ | $D_S$ |
> > > > |---|---|---|---|---|---|---|
> > > > | R34 | 22 | 73.3 | 68.6 | 64.5 | 36.8 | 2.1 |
> > > > | R101 | 45 | 77.4 | 69.1 | 62.9 | 34.0 | 1.1 |
> > > > | ViT-L | 304 |  **79.3**  |  **72.8**  | 67.7 | 41.8 | 1.1 |
> > > > | CLIP-B/32 | 151 | 60.2 | 58.4 | 56.4 | 52.3 | 14.3 |
> > > > | CLIP-L/14 | 428 | 68.1 | 67.4 | 66.7 | 65.6 | 49.1 |
> > > > | R34N (ours) | 22 | 73.2 |  **72.8**  |  **71.7**  |  **68.1**  |  **67.0**  |
> > > >
> > > > Despite CLIP’s broader training, it suffers a significant drop under intensity corruptions, while our much smaller and more narrowly trained model remains robust—even to perturbations it never saw.  From this perspective, the comparison is informative: it highlights that large-scale pretraining does not automatically confer robustness to low-level corruptions, and that classical principles like filter normalization can still play a valuable role.
> > > >
> > > > We consider our comparison with CLIP is not unfair as our goal is not to claim general superiority over CLIP, but to illustrate an aspect where even foundation models may benefit from architectural improvements. Filtering-based normalization remains relevant, particularly for CNN-based models such as CvT [1] and ConvNeXt [2].
> > > >
> > > > We will revise the paper to reflect this nuanced comparison and clarify that our claims are specific to robustness.
> > > >
> > > > [1] Wu, et al. CvT: Introducing Convolutions to Vision Transformers. ICCV 2021.
> > > >
> > > > [2] Liu, et al. A ConvNet for the 2020s. CVPR 2022.

---

### Note · Authors · 2025-08-14

**Regarding the comparison with CLIP**, we respectfully disagree with Reviewer j6NF’s continued characterization of it as “unfair” or “somewhat misleading.”

If anything, the comparison is conservative and disadvantages our method. On fairness, we emphasize two key points:

1. **Semantic generalization is not our goal**, and our model is not trained for that; we make no claim that R34N competes with CLIP in broad semantics.

2. **Our evaluation targets robustness to natural intensity corruptions**. R34N was trained solely on ImageNet-1k without any atmospheric augmentations, while CLIP was trained on 400M diverse images and likely encountered many such degradations (evidence detailed below). Yet R34N outperforms CLIP by a wide margin. This is a fair, focused, and meaningful comparison.

As shown in Table 1, our findings are consistent and decisive:

1. On the original test set $D$, **specialist models like R34N and R34** outperform generalist models like CLIP-B (73.2%, 73.3% vs. 60.2%), reflecting their domain-specific focus.

2. On corrupted sets such as $D_S$, **specialist baselines collapse** (e.g., R34 drops from 73.3% to 2.1%), while CLIP-B drops but to a higher accuracy (from 60% to 14.3%), consistent with the common belief on how large-scale pretraining offers some robustness through scale and diversity of the training data.

3. **R34N, trained on the same ImageNet-1k data as R34 but without any augmentation**, achieves 67.0% on $D_S$, far surpassing both R34 and CLIP-B, confirming that the robustness gain comes from the architecture itself, not from exposure to corruptions or tradeoffs on original clean performance.

These results lead to two major takeaways:

1. **The robustness comparison with CLIP is fair and square**. Both models face the same unseen corruptions at test time. That R34N, trained with far less data and no augmentations, decisively beats CLIP on robustness is a valid and valuable result.

2. **Filter normalization offers robustness gains that scale alone does not**. The strong generalization of R34N, without seeing any intensity corruption during training, demonstrates that classical vision principles are not only relevant but essential for achieving state-of-the-art performance, even in the foundation model era.

We will revise the manuscript to clarify these points and frame the CLIP comparison more explicitly in this context.

The remaining concerns were addressed or cleared up as indicated in the reviewers' final comments.

---

### Decision · Program_Chairs · 2025-09-17

**Decision:**

Accept (poster)

**Comment:**

This paper proposes to normalize the convolution filter in deep neural networks (including ConvNets and ViT) motivated by classical image processing. They demonstrated that normalizing the filters benefits the learning of the model. Overall, the reviewers are supportive of the paper. During the discussion period, reviewers requested several additional comparisons and discussed whether the compared baselines are fair. The authors have promised to address and add these discussions. Finally, the AC would like to point out that “atmosphere-equivariant” is not formally defined. Strictly speaking, equivariance is defined with respect to a group action, which isn’t explicitly stated. The AC understands that some works use a less formal usage of "equivariance". It is better to be clear.